# Styxl2 regulates de novo sarcomere assembly by binding to non-muscle myosin IIs and promoting their degradation

Xianwei Chen[1†], Yanfeng Li[1†], Jin Xu[1], Yong Cui[2], Qian Wu[3], Haidi Yin[3], Yuying Li[4], Chuan Gao[1], Liwen Jiang[2], Huating Wang[4], Zilong Wen[1], Zhongping Yao[3], Zhenguo Wu[1]*

[1]Division of Life Science, Hong Kong University of Science & Technology, Hong Kong, China; [2]School of Life Sciences, Chinese University of Hong Kong, Hong Kong, China; [3]Department of Applied Biology and Chemical Technology, Hong Kong Polytechnic University, Hong Kong, China; [4]Department of Orthopaedics and Traumatology, Li Ka Shing Institute of Health Sciences, Chinese University of Hong Kong, Hong Kong, China

*For correspondence:
bczgwu@ust.hk

†These authors contributed equally to this work

Competing interest: The authors declare that no competing interests exist.

**Abstract** Styxl2, a poorly characterized pseudophosphatase, was identified as a transcriptional target of the Jak1-Stat1 pathway during myoblast differentiation in culture. Styxl2 is specifically expressed in vertebrate striated muscles. By gene knockdown in zebrafish or genetic knockout in mice, we found that Styxl2 plays an essential role in maintaining sarcomere integrity in developing muscles. To further reveal the functions of Styxl2 in adult muscles, we generated two inducible knockout mouse models: one with *Styxl2* being deleted in mature myofibers to assess its role in sarcomere maintenance, and the other in adult muscle satellite cells (MuSCs) to assess its role in de novo sarcomere assembly. We find that Styxl2 is not required for sarcomere maintenance but functions in de novo sarcomere assembly during injury-induced muscle regeneration. Mechanistically, Styxl2 interacts with non-muscle myosin IIs, enhances their ubiquitination, and targets them for autophagy-dependent degradation. Without Styxl2, the degradation of non-muscle myosin IIs is delayed, which leads to defective sarcomere assembly and force generation. Thus, Styxl2 promotes de novo sarcomere assembly by interacting with non-muscle myosin IIs and facilitating their autophagic degradation.

## eLife assessment

This paper presents an **important** finding: that Styxl2, a poorly characterized pseudo-phosphatase, plays a role in the sarcomere assembly by promoting the degradation of non-muscle myosins. The genetic evidence supporting the conclusions is **compelling**, although future work will be needed to elucidate the functional role and biochemical mechanism of autophagic degradation of non-muscle myosins. This work will be of interest to biologists studying muscle development, cell biology, and proteolysis.

## Introduction

Striated muscles, including skeletal and cardiac muscles of vertebrates and some muscles of invertebrates, are unique in that they contain highly-organized, multi-protein contractile structures called

myofibrils in the cytosol of myofibers or cardiomyocytes (*Clark et al., 2002*; *Mukund and Subramaniam, 2020*; *Sparrow and Schöck, 2009*). Sarcomeres are the smallest repeating structural units of myofibrils responsible for muscle contraction (*Avellaneda et al., 2021*; *Squire, 2005*). A sarcomere is flanked by Z-discs (or Z lines) with multiple actin filaments (or thin filaments) anchored to the Z-discs in parallel and the interdigitating bipolar myosin bundles (or thick filaments) anchored to the M-line in the middle of the sarcomere (*Squire, 2005*). As a key component of the Z-disc, an anti-parallel α-actinin dimer directly binds to two actin filaments in opposite directions (*Squire, 2005*). A sarcomere can be further divided into the lighter I (isotropic)-band and the darker A (anisotropic)-band based on their microscopic appearances with the I-band containing actin filaments only and the A-band containing both the myosin bundles and partially overlapping actin filaments (*Craig and Padrón, 2004*; *Squire, 2005*). Mutations in sarcomeric proteins are linked to congenital human cardiac and skeletal muscle diseases (*Bönnemann and Laing, 2004*; *Parker et al., 2020*; *Parker and Peckham, 2020*). Although the overall structures of sarcomeres are well-established and the protein components of sarcomeres are largely known, it remains controversial how sarcomeres correctly assemble in vivo. Several different but not mutually exclusive models including the template model, the stitching model, and the premyofibril model have been proposed (*Martin and Kirk, 2020*; *Rui et al., 2010*; *Sanger et al., 2006*). The template model proposes that actin stress fibers serve as an early template to recruit other sarcomeric components including Z-disc proteins and the thin and thick filaments (*Dlugosz et al., 1984*). The stitching model or the titin model proposes that the actin filaments held by the Z-disc (i.e. the I-Z-I complexes) and the myosin filaments first assemble independently, and they are then stitched together to form intact sarcomeres with the help of titin as titin is known to simultaneously interact with Z-disc proteins via its N-terminal region and M-line proteins via its C-terminal region (*Holtzer et al., 1997*; *Lu et al., 1992*). The premyofibril model states that mini-sarcomeres consisting of actin filaments, α-actinin-containing Z bodies, and non-muscle myosin IIs first assemble near cell periphery. Then, non-muscle myosin IIs are replaced by muscle myosin II during the transition from premyofibrils to nascent myofibrils and finally to mature myofibrils (*Rhee et al., 1994*; *Sanger et al., 2006*). Each of these models is supported by some experimental observations but may be incompatible with others (*Chopra et al., 2018*; *Fenix et al., 2018*; *Rui et al., 2010*). In addition, mechanical tension generated by attachment of myofibers to tendon, myofiber bundling, and formation of mini-sarcomeres is also found to promote sarcomere assembly in both fly and mammalian muscle cells (*Chopra et al., 2018*; *Loison et al., 2018*; *Luis and Schnorrer, 2021*; *Mao et al., 2022*; *Weitkunat et al., 2014*).

Protein phosphatases are a family of enzymes that catalyze the removal of the phosphate group from covalently-linked substrates including proteins, lipids, nucleic acids, and polysaccharides (*Alonso et al., 2004*; *Tonks, 2006*; *Weinfeld et al., 2011*; *Worby et al., 2006*). They antagonize the actions of kinases and regulate various biological processes including tissue development, cell proliferation, differentiation, and cell migration. Interestingly, some of the protein phosphatases are found to be catalytically inactive due to the absence of essential amino acid residues such as the catalytic cysteine in an otherwise well-conserved phosphatase domain (*Chen et al., 2017*; *Reiterer et al., 2020*). These confirmed or putative catalytically inactive phosphatases are also called pseudophosphatases. Although catalytically inactive, several pseudophosphatases are found to play important roles in various organisms (*Azzedine et al., 2003*; *Cheng et al., 2009*; *Hinton, 2020*; *Reiterer et al., 2017*). For example, Styx and Sbf1, both members of the pseudophosphatase family, were found to regulate sperm development. Loss of Styx or Sbf1 resulted in male infertility in mice (*Firestein et al., 2002*; *Wishart and Dixon, 2002*).

In our current study, we identified Styxl2 (previously named as Dusp27), a poorly characterized pseudophosphatase, as a new regulator involved in sarcomere assembly in both zebrafish and mice. The original name of the gene and its protein product (i.e. Dusp27) has caused confusion in the literature due to the fact that the same name was once used for Dupd1 (now re-named as Dusp29), a catalytically active phosphatase (*Friedberg et al., 2007*). Recently, it was proposed that Dusp27 be re-named as Serine/threonine/tyrosine-interacting-like 2 (Styxl2) (*Cooper and Waddell, 2020*). To avoid further confusion, we will use Styxl2 hereafter. Styxl2 is specifically expressed in vertebrate striated muscles. Our work in both zebrafish and mouse models shows that it plays an important role in de novo sarcomere assembly instead of sarcomere maintenance. Mechanistically, we found that Styxl2 directly interacts with non-muscle myosin IIs that are involved in the early stage of sarcomere assembly and subsequently replaced by muscle myosin IIs in mature sarcomeres (*Wang et al., 2018*). Styxl2

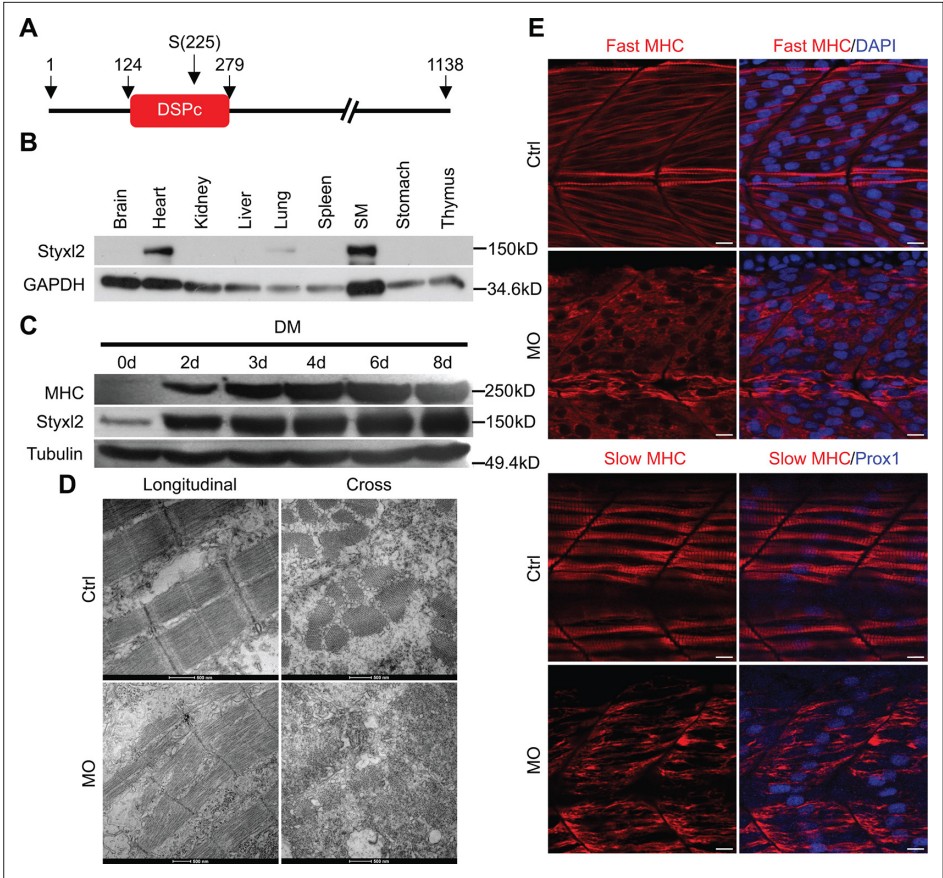

**Figure 1.** Styxl2 regulates sarcomere integrity during zebrafish muscle development. (**A**) A schematic of mouse Styxl2 protein. DSPc (red): dual-specificity phosphatase, catalytic domain. The numbers denote the positions of various amino acids in mouse Styxl2 including Ser (S) –225. (**B**) The expression of Styxl2 protein in different mouse tissues. 100 μg of soluble mouse tissue lysates were subjected to Western blot analysis. GAPDH: glyceraldehyde-3-phosphate dehydrogenase. SM: skeletal muscles. (**C**) The expression of Styxl2 protein in C2C12 myoblasts before and after differentiation in culture. MHC: myosin heavy chain. (**D**) Zebrafish zygotes were either mock-injected or injected with a Styxl2-specific morpholino (MO). Sarcomeres in muscles at 48 hpf were revealed by transmission electron microscopy (TEM). Representative images were shown. Scale bar: 500 nm. (**E**) Sarcomeres in fast and slow muscles of zebrafish embryos at 24 hpf were revealed by immunostaining using specific antibodies as indicated. Ctrl: control (non-treated); MO: *Styxl2*-morpholino-treated. The nuclei of slow and fast muscle fibers were counter-stained with an anti-Prox1 antibody and DAPI, respectively. Scale bar: 10 μm.

The online version of this article includes the following source data and figure supplement(s) for figure 1:

**Source data 1.** Original scans for the Western blot analysis in *Figure 1B* (anti-Styxl2 and anti-GAPDH).

**Source data 2.** A PDF file showing original scans of the relevant Western blot analysis in *Figure 1B* (anti-Styxl2 and anti-GAPDH) with highlighted bands and sample labels.

**Source data 3.** Original scans for the Western blot analysis in *Figure 1C* (anti-MHC, anti-Styxl2, and anti-Tubulin).

**Source data 4.** A PDF file showing original scans of the relevant Western blot analysis in *Figure 1C* (anti-MHC, anti-Styxl2, and anti-Tubulin) with highlighted bands and sample labels.

**Figure supplement 1.** Styxl2 was downstream target of Jak1-Stat1 pathway.

**Figure supplement 1—source data 1.** Original scans for the Western blot analysis in *Figure 1—figure supplement 1C* (anti-Jak1, anti-Stat1, anti-Styxl2, and anti-Tubulin).

**Figure supplement 1—source data 2.** A PDF file showing original scans of the relevant Western blot analysis in *Figure 1—figure supplement 1C* (anti-Jak1, anti-Stat1, anti-Styxl2, and anti-Tubulin) with highlighted bands and sample labels.

**Figure supplement 1—source data 3.** Original scans for the Western blot analysis in *Figure 1—figure supplement 1D* (anti-Styxl2).

*Figure 1 continued on next page*

*Figure 1 continued*

**Figure supplement 1—source data 4.** Original scans for the Western blot analysis in *Figure 1—figure supplement 1D and E* (anti-Tubulin in D, anti-Styxl2, and anti-Tubulin in E).

**Figure supplement 1—source data 5.** A PDF file showing original scans of the relevant Western blot analysis in *Figure 1—figure supplement 1D and E* (anti-Styxl2 and anti-Tubulin in D, anti-Styxl2, and anti-Tubulin in E) with highlighted bands and sample labels.

promotes ubiquitination and autophagy-dependent degradation of non-muscle myosin IIs, which ultimately facilitates sarcomere assembly in vivo.

## Results

### *Styxl2* encodes a striated muscle-specific pseudophosphatase and is a transcriptional target of the Jak1-Stat1 pathway in myoblasts

We previously reported that the Jak1-Stat1 pathway plays an important role in myogenic differentiation in cultured myoblasts (*Sun et al., 2007*). However, it remains unclear what genes are targeted by the pathway. To examine the changes in transcriptomes, we conducted a microarray analysis in C2C12 myoblasts: cells were first transfected with either a control siRNA or specific siRNAs targeting *Jak1* or *Stat1* and the cells were harvested at two different time points: (1) after 24 hr of growth in the growth medium (GM 24 hr, the proliferation stage); (2) after 12 hr in the differentiation medium following 24 hr of growth in the GM (DM 12 hr, the early differentiation stage). The total mRNA was extracted and subjected to microarray analysis. Among differentially expressed genes (DEGs) in cells transfected with *Jak1*-siRNA and *Stat1*-siRNA, many were shared, which was expected as both Jak1 and Stat1 function in the same pathway. Some representative targets shared by Jak1 and Stat1 were shown (*Table 1*). Consistent with our previous findings (*Sun et al., 2007*), *myogenin* gene was negatively regulated by both Jak1 and Stat1. Among the genes targeted by both Jak1 and Stat1, *serine/threonine/tyrosine-interacting-like 2* (*Styxl2*), previously also named as *dual-specificity phosphatase 27 (putative)* (*Dusp27*), attracted our attention: it is a poorly characterized gene with only one published report in zebrafish so far (*Fero et al., 2014*). It encodes a member of the dual-specificity phosphatase family with a conserved dual-specificity phosphatase catalytic (DSPc) domain located at its N-terminus (*Figure 1A*). However, Styxl2 is predicted to be catalytically inactive due to substitution of a highly-conserved catalytic cysteine by serine in its DSPc domain (*Figure 1—figure supplement 1A*), thus making it a member of the pseudophosphatase subfamily (*Hinton, 2019*; *Kharitidi et al., 2014*). By both RT-qPCR and Western blot, we confirmed that both the mRNA and protein levels of *Styxl2* were induced by siRNAs against *Jak1* or *Stat1* in C2C12 myoblasts, with the extent of induction being more obvious during early differentiation (*Figure 1—figure supplement 1B and C*). Consistently, treatment of C2C12 myoblasts with leukemia inhibitory factor (LIF) or oncostatin M (OSM), two cytokines that are known to potently activate the Jak1-Stat1 pathway in myoblasts (*Sun et al., 2007*; *Xiao et al., 2011*), indeed led to a gradual decrease of Styxl2 protein (*Figure 1—figure supplement 1D and E*). By Western blot, we found that *Styxl2* has a very limited tissue distribution pattern: it is specifically expressed in striated muscles including the heart and skeletal muscles (*Figure 1B*). Consistently, the expression of *Styxl2* mRNA is confined to the heart and somites of early mouse embryos (*Figure 1—figure supplement 1F*). In C2C12 myoblasts, Styxl2 protein was already present in proliferating myoblasts but was further induced during myogenic differentiation (*Figure 1C*). We also examined the subcellular localization of Styxl2 in myoblasts. As our home-made polyclonal antibody is not suitable for immunostaining of the endogenous Styxl2, we examined subcellular localization of Flag- or HA-tagged Styxl2 instead in C2C12 cells by immunostaining. We found that both Flag- and HA-tagged Styxl2 were predominantly localized in the cytoplasm (*Figure 1—figure supplement 1G*). Collectively, Styxl2 is a cytoplasmic protein that is specifically expressed in striated muscles.

### Knockdown of *Styxl2* in zebrafish leads to severe defects in sarcomere structure

An earlier study by *Fero et al., 2014* in zebrafish showed that Styxl2 regulates the integrity of sarcomeres. We also confirmed this finding in zebrafish using a morpholino oligo (MO) that blocks the

**Table 1.** Candidate genes regulated by both Jak1 and Stat1.

Selected candidate genes targeted by both si-Jak1 and si-Stat1 were shown. The fold change was determined by the relative levels of a gene in cells treated with si-Jak1 or si-Stat1 over that in control cells. Genes with fold change >1.5 for si-Jak1 and >2 for si-Stat1 were listed in this table. Genes with positive fold change are up-regulated (positive value), while those with negative fold change are down-regulated (negative value). P Stage and D Stage denote the proliferation stage (growth medium. GM 24 hr) and early differentiation stage (differentiation medium, DM 12 hr), respectively.

| Probe set ID | Gene | Fold change | | | |
| | | P Stage | | D Stage | |
| | | siJak1 | siStat1 | siJak1 | siStat1 |
|---|---|---|---|---|---|
| 1417889_at | apolipoprotein B editing complex 2 | 1.53 | 4.43 | 2.43 | 4.07 |
| 1419391_at | myogenin | 4.29 | 9.99 | 2.62 | 4.26 |
| 1422088_at | v-myc myelocytomatosis viral oncogene homolog 1 | 2.12 | 3.9 | 1.88 | 3.68 |
| 1422606_at | C1q and tumor necrosis factor related protein 3 | 2.23 | 13.4 | 1.68 | 5.45 |
| 1426971_at | ubiquitin-activating enzyme E1-like | 3.53 | 7.05 | 2.42 | 3.03 |
| 1427306_at | ryanodine receptor 1 | 2.11 | 6 | 2.71 | 4.52 |
| 1449178_at | PDZ and LIM domain 3 | 3.45 | 12.3 | 2.94 | 7 |
| 1451453_at | death-associated kinase 2 | 1.52 | 3.14 | 1.96 | 3.81 |
| 1452520_a_at | cholinergic receptor | 6.76 | 15.5 | 2.6 | 4.13 |
| 1429223_a_at | hemochromatosis type 2 (juvenile) (human homolog) | 2.16 | 28.8 | 2 | 7.66 |
| 1429459_at | sema domain | 2.77 | 4.43 | 2.27 | 5.58 |
| 1435828_at | avian musculoaponeurotic fibrosarcoma (v-maf) AS42 oncogene homolog | 1.96 | 3.38 | 1.63 | 3.27 |
| 1439658_at | leiomodin 3 (fetal) | 1.91 | 17.9 | 2.83 | 18.8 |
| 1439746_at | dual specificity phosphatase 27 (putative) (dusp27) | 1.82 | 4.78 | 1.73 | 2.36 |
| 1444494_at | kelch repeat and BTB (POZ) domain containing 10 | 2.46 | 9.63 | 2.79 | 6.38 |
| 1421426_at | Hedgehog-interacting protein | −2.04 | −3.55 | −1.62 | −6.65 |
| 1435438_at | SRY-box containing gene 8 (sox8) | 1.85 | 4.01 | 1.51 | 5.7 |

translation initiation of *Styxl2*. The effectiveness of the MO was verified using a transgene encoding green fluorescence protein (GFP) fused in-frame with the same sequence of *Styxl2* (i.e. the first 25 nucleotides of the coding region) targeted by the MO: when zebrafish zygotes were microinjected with the transgenes with or without the MO, the expression of GFP was suppressed only when the MO was co-injected (*Figure 1—figure supplement 1H*). We then injected zebrafish zygotes with or without the *Styxl2*-MO and examined the structures of skeletal muscles by immunostaining at 24 hr after fertilization (hpf). Consistent with the report by *Fero et al., 2014*, we found that sarcomeres in both fast and slow muscles were severely disrupted when *Styxl2* was knocked down with the MO (*Figure 1E*). Moreover, electron microscopy analysis further revealed the disrupted sarcomere structures in zebrafish muscles injected with the MO (*Figure 1D*).

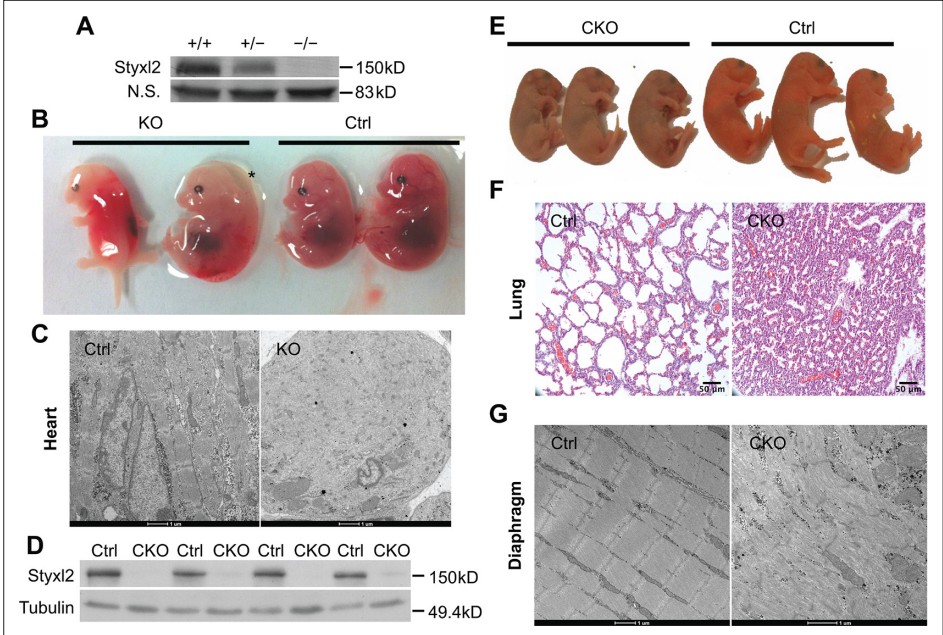

**Figure 2.** Conditional deletion of mouse *Styxl2* leads to defective sarcomeres in striated muscles. (**A**) The expression of Styxl2 protein in skeletal muscles of wild-type (+/+), heterozygous (+/−), and *Styxl2* knockout (KO) (−/−) (driven by *EIIA*-Cre) mice. Soluble tissue lysates from limbs of E14.5 mouse embryos were subjected to Western blot analysis. N.S.: non-specific. (**B**) E14.5 embryos of control (Ctrl) and *EIIA*-Cre; *Styxl2^{f/f}* KO mice. *: edema was found in some *Styxl2* KO foetuses. (**C**) The sarcomere structures of cardiomyocytes in the heart of E14.5 embryos from control and *EIIA*-Cre; *Styxl2^{f/f}* KO mice were revealed by transmission electron microscopy (TEM). Scale bar: 1 µm. (**D**) The expression of Styxl2 protein in skeletal muscles of control (Ctrl) and *Styxl2* CKO P1 mice. (**E**) The new-born pups (P1) from control (Ctrl) and *Pax7^{Cre/+}*; *Styxl2^{f/f}* KO mice (CKO). (**F**) Lung sections of control and Styxl2 CKO P1 mice. Scale bar: 50 µm. (**G**) The sarcomere structure of diaphragm muscles from control (Ctrl) and *Styxl2* conditional KO (CKO) P1 mice revealed by TEM. Scale bar: 1 µm.

The online version of this article includes the following source data and figure supplement(s) for figure 2:

**Source data 1.** Original scans for the Western blot analysis in *Figure 2A* (anti-Styxl2 and N.S.).

**Source data 2.** A PDF file showing original scans of the relevant Western blot analysis in *Figure 2A* (anti-Styxl2 and N.S.) with highlighted bands and sample labels.

**Source data 3.** Original scans for the Western blot analysis in *Figure 2D* (anti-Styxl2 and anti-Tubulin).

**Source data 4.** A PDF file showing original scans of the relevant Western blot analysis in *Figure 2D* (anti-Styxl2 and anti-Tubulin) with highlighted bands and sample labels.

**Figure supplement 1.** Both germline and conditional deletion of *Styxl2* caused lethality.

## Conditional deletion of *Styxl2* leads to neonatal lethality in mutant mice

To further explore the role of Styxl2 in mammals, we generated *Styxl2* floxed mice with two LoxP sites flanking exons 3~5 of the *Styxl2* gene (*Figure 2—figure supplement 1A*). Exons 4 and 5 encode a part of the DSPc domain of Styxl2. We first generated germline *Styxl2* knockout (KO) mice using EIIA-Cre mice (*Lakso et al., 1996*). Western blot analysis showed that Styxl2 protein was totally absent in KO mice (*Figure 2A*). In multiple litters, no live homozygous KO pups were found after birth (*Figure 2— figure supplement 1B*). By examining developing embryos of different ages, we found that *Styxl2* KO embryos died between E13.5 and E15.5 (*Figure 2—figure supplement 1C*). In some dead *Styxl2* KO fetuses, edema was found (*Figure 2B*), suggesting potential problems with the heart (*Conway et al., 2003*). This is consistent with the fact that Styxl2 is also highly expressed in cardiac muscles as well. The electron microscopy analysis of the heart muscles from E14.5 embryos revealed that the sarcomeric structures of cardiac muscles from *Styxl2* KO embryos were severely disrupted (*Figure 2C*). To focus on the functions of Styxl2 in skeletal muscles, we generated conditional *Styxl2* KO (CKO) mice by crossing *Styxl2^{f/f}* mice with *Pax7^{Cre/+}* mice (*Keller et al., 2004*; *Figure 2—figure supplement 1D*).

We found that *Styxl2* CKO pups were born at the expected Mendelian ratio but all the homozygous mutant pups died in a few hours after birth (*Figure 2D and E* and *Figure 2—figure supplement 1D*). As Pax7 is also expressed in other non-muscle cell types including neurons during embryo development, to be certain that embryonic lethality of *Pax7^Cre/+^; Styxl2^f/f^* mice was indeed caused by severe defects in skeletal muscles, we also specifically deleted *Styxl2* in skeletal muscles using a *Myf5^Cre^* line in which the *Cre* gene was inserted in the *Myf5* locus (*Tallquist et al., 2000*). Consistently, all skeletal muscle-specific *Styxl2* KO pups died within one day after birth (*Figure 2—figure supplement 1D*).

As the skin of *Styxl2* CKO pups appeared cyanotic, we speculated that the new-born mutant mice had lung defects that led to respiratory distress and perinatal death. The lungs of the CKO pups were examined and we found that the pulmonary alveoli indeed failed to open (*Figure 2F*), which was likely caused by defective diaphragm muscle as defects in a number of myogenic genes exhibited similar perinatal lethality (*Hasty et al., 1993*; *Rudnicki et al., 1993*). Electron microscopy analysis confirmed that the sarcomere structure was severely disrupted in diaphragm muscle of the mutant mice (*Figure 2G*). Consistently, electron microscopy revealed that the sarcomere structure of the limb muscle was also severely disrupted (*Figure 2—figure supplement 1E*). Notably, some sarcomeres were still present in the skeletal muscles of the CKO mice (*Figure 2—figure supplement 1E*), suggesting that Styxl2 is important but not absolutely required for the formation of the sarcomeres. Therefore, Styxl2 plays an important role in the development of striated muscles, including the skeletal muscle and the heart muscle.

## Styxl2 is dispensable for the maintenance of mature sarcomeres but critical for de novo sarcomere assembly in adult skeletal muscles

Although our data above indicated that Styxl2 plays an important role in regulating sarcomere formation during embryonic skeletal muscle development, it remained unclear whether it was necessary for the maintenance of the sarcomere structure in adult skeletal muscles. To address this question, we generated a tamoxifen (TMX) inducible myofiber-specific *Styxl2* knockout mouse model (also named *Styxl2* MF-iKO hereafter) using Tg (HSA-MerCreMer) (*McCarthy et al., 2012*). In this model, *Styxl2* was efficiently deleted in mature skeletal muscles of adult mice (*Figure 3A and B*). We directly assessed the contractile functions of Styxl2-null muscles. Three weeks after the last TMX injection (*Figure 3A*), the gastrocnemius muscle was isolated and the twitch and tetanic force generated by the muscle were measured (*Guo et al., 2016*). No obvious difference in twitch or tetanic force was observed between the control and *Styxl2* MF-iKO muscles (*Figure 3C and D*), suggesting that Styxl2 is dispensable for the maintenance of existing sarcomeres in adult skeletal muscles. Next, we assessed whether Styxl2 played a role in the de novo formation of nascent sarcomeres in regenerating muscles of adult mice. To do so, we generated another tamoxifen-inducible, MuSC-specific *Styxl2* KO mouse model using *Pax7^CreERT2/CreERT2^; Styxl2^f/f^* (also named *Styxl2* SC-iKO mice hereafter) (*Murphy et al., 2011*). *Styxl2* was first deleted in muscle satellite cells (MuSCs) of adult mice according to the scheme in *Figure 3E*, followed by acute injury to the hindlimb skeletal muscles with Cardiotoxin (CTX) (*Figure 3E*). As injured muscles are repaired by MuSCs, the nascent sarcomeres in regenerating myofibers of *Styxl2* SC-iKO mice would be formed in the absence of Styxl2 (*Figure 3F*). At 30 days post-injury, a time point when injured muscles are known to be fully regenerated (*Dumont et al., 2015*), the gastrocnemius muscles were isolated and functionally tested in vitro. Both the twitch and tetanic force were measured separately. Compared to muscles from the control mice, the twitch force generated by muscles from *Styxl2* SC-iKO mice was decreased by ~30% (*Figure 3G*), while the tetanic force generated by muscles from the mutant mice was not affected (*Figure 3H*). This suggested that sarcomeres formed de novo in regenerating myofibers from Styxl2-null MuSCs were partially defective. When we examined sarcomeres by electron microscopy, we noticed that the length of individual sarcomeres from *Styxl2* SC-iKO mice was shorter in regenerated muscles (*Figure 3I and J*). Thus, our data showed that Styxl2 functions in de novo sarcomere assembly in regenerating muscles from adult mice.

## Styxl2 directly binds to non-muscle myosin II (NM II) in myoblasts

To understand how Styxl2 functions to regulate sarcomere formation, we employed BioID, a proximity-dependent protein labelling method (*Roux et al., 2012*; *Roux et al., 2018*), to search for binding partners of Styxl2. The full-length Styxl2 was fused in-frame with BirA*, a mutant form of biotin ligase BirA (*Figure 4A*). The fused gene encoding Styxl2-BirA*-HA was stably expressed

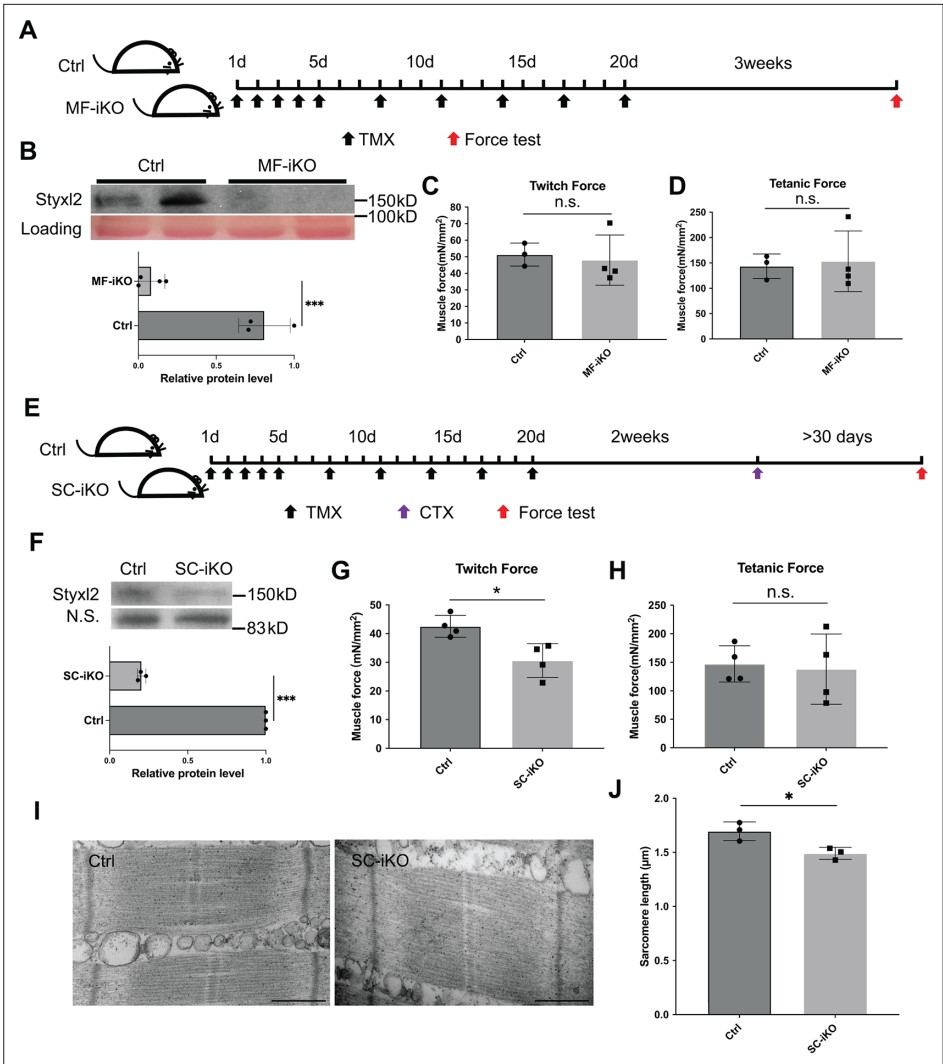

**Figure 3.** Styxl2 is not required for sarcomere maintenance but involved in de novo sarcomere assembly in adult muscles. (**A**) The schematic to assess the role of Styxl2 in sarcomere maintenance in adult muscles. TMX: tamoxifen. (**B**) Styxl2 protein levels in control (Ctrl) and myofiber-specific *Styxl2* knockout (MF-iKO) mouse gastrocnemius muscles used for force test. The band intensity of Styxl2 was quantified by ImageJ. (**C, D**) The twitch and tetanic force generated by gastrocnemius muscles of three Ctrl and four MF-iKO mice were measured. In (**A–D**), Ctrl: *Styxl2^{f/f}* mice; MF-iKO: *Tg: HSA-MerCreMer; Styxl2^{f/f}* mice. (**E**) The schematic to assess the role of Styxl2 in de novo sarcomere assembly in adult muscles. CTX: cardiotoxin. (**F**) Styxl2 protein levels in regenerated gastrocnemius muscles used for force test. N.S.: non-specific. The band intensity of Styxl2 was quantified. (**G, H**) Twitch and tetanic force by regenerated gastrocnemius muscles were measured from four pairs of control and MuSC-specific *Styxl2* KO (SC-iKO) mice. (**I**) Transmission electron microscopy (TEM) images of the muscle used for force generation test. Scale bar: 500 nm. (**J**) Quantification of sarcomere length in (**I**). TEM images from three pairs of mice were examined and the length of 3–5 sarcomeres from each mouse was measured. In (**E–J**), Ctrl: *Pax7^{CreERT2/CreERT2}* mice; SC-iKO: *Pax7^{CreERT2/CreERT2}; Styxl2^{f/f}* mice. n.s.: not significant. *p-value <0.05. ***p-value <0.001.

The online version of this article includes the following source data for figure 3:

**Source data 1.** Original scans for the Western blot analysis in *Figure 3B* (anti-Styxl2 and Loading).

**Source data 2.** A PDF file showing original scans of the relevant Western blot analysis in *Figure 3B* (anti-Styxl2 and Loading) with highlighted bands and sample labels.

**Source data 3.** Original scans for the Western blot analysis in *Figure 3F* (anti-Styxl2 and N.S.).

**Source data 4.** A PDF file showing original scans of the relevant Western blot analysis in *Figure 3F* (anti-Styxl2 and N.S.) with highlighted bands and sample labels.

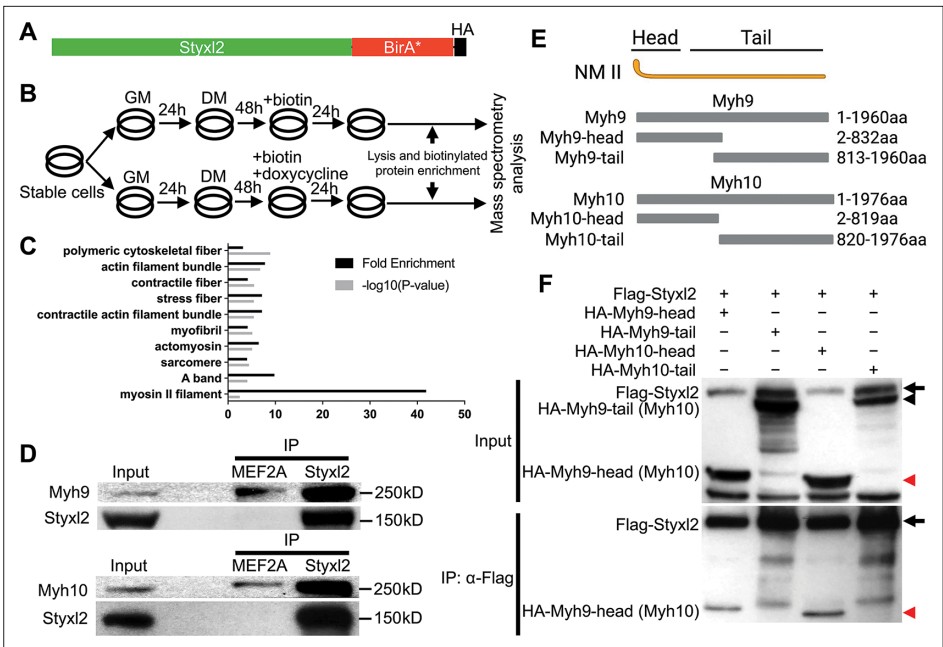

**Figure 4.** Styxl2 binds to non-muscle myosin IIs. (**A**) The schematic of the fusion protein Styxl2-BirA*-HA. (**B**) The workflow of sample preparation for BioID. (**C**) Enriched (with Enrichment score >1) biotinylated proteins in cells expressing Styxl2-BirA*-HA relative to that in control cells were identified by mass spectrometry followed by Gene Ontology (GO) analysis. Myofibril-related terms were shown. (**D**) C2C12 cells were harvested after 2 days in differentiation medium (DM) and the soluble whole cell lysates were subjected to immunoprecipitation (IP) using either an anti-MEF2A (negative control) or an anti-Styxl2 antibody. The endogenous Myh9 or Myh10 was detected by Western blot. (**E**) The schematic shows the domains of non-muscle myosin IIs (NM II). (**F**) HEK 293T cells were transfected with Flag-Styxl2 (black arrow) together with HA-NM II-head (red arrowheads) or HA-NM II-tail (black arrowheads). The soluble whole cell lysates were subjected to IP with an anti-Flag antibody and the co-immunoprecipitated proteins were detected by Western blot.

The online version of this article includes the following source data and figure supplement(s) for figure 4:

**Source data 1.** Original scans for the Western blot analysis of input and IP in *Figure 4D* (anti-Styxl2, anti-Myh9, and anti-Myh10).

**Source data 2.** A PDF file showing original scans of the relevant Western blot analysis in *Figure 4D* (anti-Styxl2, anti-Myh9, and anti-Myh10) with highlighted bands and sample labels.

**Source data 3.** Original scans for the Western blot analysis of input and IP in *Figure 4F* (anti-Flag and anti-HA).

**Source data 4.** A PDF file showing original scans of the relevant Western blot analysis in *Figure 4F* (anti-Flag and anti-HA) with highlighted bands and sample labels.

**Figure supplement 1.** Identification of Styxl2-interacting partners by BioID.

**Figure supplement 1—source data 1.** Original scans for the Western blot analysis in *Figure 4—figure supplement 1A* (Streptavidin-HRP, anti-Styxl2, and Loading).

**Figure supplement 1—source data 2.** A PDF file showing original scans of the relevant Western blot analysis in *Figure 4—figure supplement 1A* (Streptavidin-HRP, anti-Styxl2, and Loading) with highlighted bands and sample labels.

**Figure supplement 1—source data 3.** An Excel file showing raw data of iTRAQ-based mass spectrometry analysis for potential Styxl2 interactors in *Figure 4—figure supplement 1B*.

---

in C2C12 myoblasts using the Tet-on system (*Das et al., 2016*; *Figure 4—figure supplement 1A*). As Styxl2 is expected to function during late myogenic differentiation, we followed the scheme in *Figure 4B* and added biotin into the culture medium with or without doxycycline (Dox) for 24 hr before harvest. By Western blot, we showed that Styxl2-BirA*-HA was strongly induced by Dox as expected (*Figure 4—figure supplement 1A*, middle panel). Importantly, more cellular proteins were biotinylated in Styxl2-BirA*-HA-expressing cells induced by Dox (*Figure 4—figure supplement 1A*, top panel). These biotinylated proteins were isolated and enriched by streptavidin beads

and analysed by iTRAQ-based mass spectrometry (*Wiese et al., 2007*). The biotinylated proteins enriched in Styxl2-BirA*-HA-expressing cells (Enrichment score >1) were further analysed by gene ontology (GO) analysis. Notably, a number of the enriched proteins were related to 'myofibril', 'sarcomere', and 'A band' (*Figure 4C*). This supports our hypothesis that Styxl2 functions to regulate sarcomere assembly. To help identify direct interaction partners of Styxl2, the log$_2$ Enrichment scores (i.e. the peak intensity ratios of the reporter ions) were plotted against the minus log$_{10}$(p.values). Those enriched proteins with the Enrichment score >1.2 and p<0.05 were highlighted in red on the plot (*Figure 4—figure supplement 1B*), and some of the enriched proteins (*Figure 4—figure supplement 1C*) were selected for further confirmation by co-immunoprecipitation (co-IP) assays using lysates from differentiated C2C12 myotubes. Among them, Myh9 and Myh10, the members of the non-muscle myosin IIs (NM IIs) (*Newell-Litwa et al., 2015*), were found to specifically interact with the endogenous Styxl2 (*Figure 4D*). Moreover, by protein truncation analysis in combination with the co-IP assays, we showed that the head domains of Myh9 and Myh10 were responsible for their interaction with Styxl2 (*Figure 4E and F*).

## Styxl2 promotes sarcomere assembly by down-regulating NM IIs

Our data above indicated that Styxl2 regulates sarcomere assembly. NM IIs were reported to participate in sarcomere assembly in several organisms (*Fenix et al., 2018*; *Sanger et al., 2009*; *Tullio et al., 1997*). However, NM IIs are not detected in mature sarcomeres and their replacement by muscle-specific myosin is a marker of skeletal muscle maturation. Therefore, we examined protein levels of Myh9 and Myh10 in both differentiating C2C12 myotubes and postnatal mouse skeletal muscles over time. In differentiating C2C12 cells, we found that the protein levels of Myh9 and Myh10 slightly increased at the beginning of myogenic differentiation, but started to decrease ~four days after myogenic differentiation, which coincided with the time when the sarcomeric myosin heavy chain started to increase (*Figure 5—figure supplement 1A*). In postnatal mouse skeletal muscles, the protein levels of Myh9 and Myh10 started to drop at P8 and became non-detectable by P30 (*Figure 5A*), which was consistent with previous reports (*Fenix et al., 2018*; *Sanger et al., 2009*). The protein levels of Styxl2 remained constant and abundant in the first month after birth but began to drop to a lower level by P65 (*Figure 5A*). When we examined the protein levels of Myh9 and Myh10 in skeletal muscles of P1 pups, we noticed an obvious increase in protein levels of Myh9 and Myh10 in *Styxl2* CKO mice (*Figure 5B and C*). We also examined protein levels of Myh9 and Myh10 in regenerating adult muscles. Using SC-iKO mice and *Pax7$^{CreERT2/CreERT2}$* control mice as described in *Figure 3E*, we first induced *Styxl2* deletion by tamoxifen in adult MuSCs followed by BaCl$_2$-induced muscle injury. At 7 dpi, the levels of Myh9 and Myh10 were comparable between control and *Styxl2* SC-iKO mice (*Figure 5D*). Interestingly, at 10 dpi, Myh9 and Myh10 were nearly undetectable in muscles from control mice, but they remained at a higher level in muscles from *Styxl2* SC-iKO mice (*Figure 5D*). Our data above suggested that loss of Styxl2 results in delayed degradation of NM II during sarcomere assembly. To further prove this is the case, we turned to the zebrafish model. It was reported that *Styxl2$^{-/-}$* zebrafish exhibited severe motility defects in response to needle touch (*Fero et al., 2014*). Consistently, we also found that *Styxl2* morphants displayed similar phenotypes: most of the *Styxl2* morphants did not respond to the needle touch (*Figure 5E*), which was caused by defective sarcomere structures in skeletal muscles (*Figure 5F*). Knowing that NM IIs are involved in the early phase of sarcomere assembly followed by their replacement with the sarcomeric myosin (*Fenix et al., 2018*; *White et al., 2014*), we suspected that delayed degradation of NM IIs was responsible for defective sarcomere structure in *Styxl2* morphants. Therefore, it was likely that simultaneous knockdown of both NM IIs and *Styxl2* may rescue the sarcomeric defects in *Styxl2* morphants. It was reported that both *Myh9b* and *Myh10* are expressed in skeletal muscles of zebrafish (*Gutzman et al., 2015*). When we knocked down *Myh9b* with a MO in *Styxl2* morphants, the embryos died very early before skeletal muscles were formed (our unpublished data). When zebrafish zygotes were co-injected with the *Styxl2*-MO and a MO targeting *Myh10* (*Figure 5—figure supplement 1B*), the percentage of fishes responding to the needle touch was obviously increased (*Figure 5E*). Consistently, the disrupted sarcomere structure caused by *Styxl2*-MO was indeed partially restored in the *Styxl2/Myh10* double-knockdown fishes (*Figure 5F*).

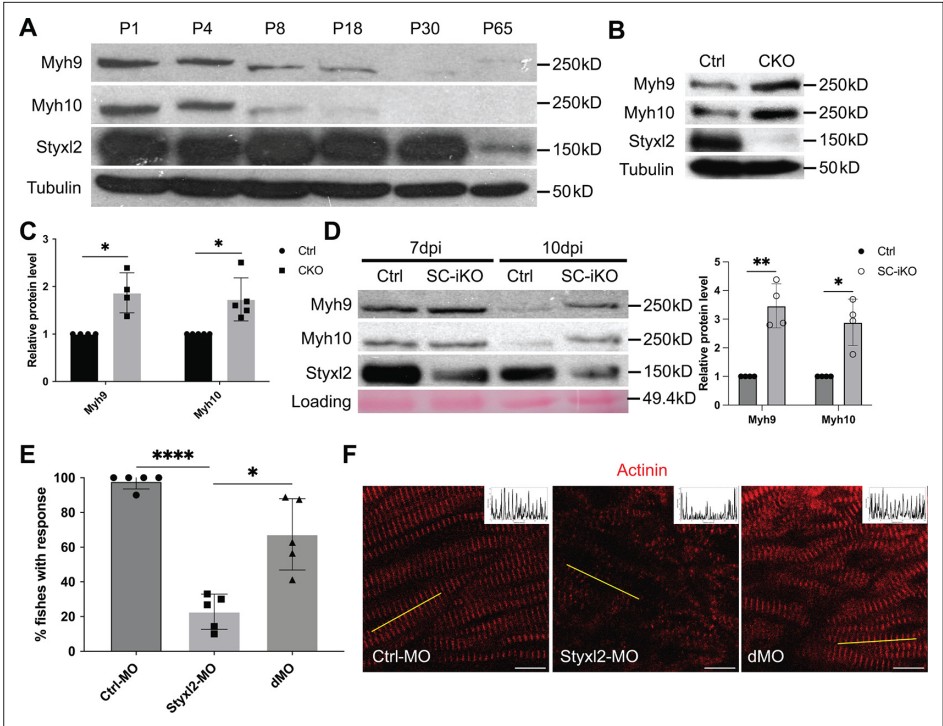

**Figure 5.** Styxl2 protein levels inversely correlates with that of non-muscle myosin IIs. (**A**) Protein levels of Styxl2 and non-muscle myosin IIs in skeletal muscles of wild-type mice of different ages were determined by Western blot. (**B**) Protein levels of Styxl2 and non-muscle myosin IIs in skeletal muscles of control and *Styxl2* conditional KO (CKO) P1 mice were determined by Western blot. (**C**) Relative ratios of non-muscle myosin II protein levels in control and *Styxl2* CKO P1 mice were quantified by ImageJ based on the data from four (for Myh9) or five (for Myh10) sets of Western blots. (**D**) Protein levels of Styxl2 and non-muscle myosin IIs in regenerating tibialis anterior muscles were determined by Western blot. The experiments were done for four times with similar results and a representative gel was shown. The protein levels of NM IIs in 10 dpi muscles revealed by Western blot were quantified in the right panel. Ctrl: *Pax7^{CreERT2/CreERT2}* mice; SC-iKO: *Pax7^{CreERT2/CreERT2}*; *Styxl2^{f/f}* mice; dpi: days post-injury. (**E**) Zebrafish zygotes were separately injected with the following morpholinos (MO): control (Ctrl-MO), *Styxl2*-MO, and mixed morpholinos targeting both *Styxl2* and *Myh10* (dMO). At 48 hpf, zebrafishes were poked by a fine needle and their responses were recorded. Quantification of responses of zebrafish to needle touch was shown. In each group, at least eight zebrafishes were tested and the percentage of zebrafishes responding to needle touch was calculated. Each dot represents one group of experiment. (**F**) Zebrafishes from each condition in (**E**) were subjected to immunostaining for α-actinin to visualize sarcomere structures. Representative images were shown. The inset at the top right corner of each image shows the actinin intensity along the yellow line. Scale bar: 10 µm. *p-value <0.05. **p-value <0.01. ****p-value <0.0001.

The online version of this article includes the following source data and figure supplement(s) for figure 5:

**Source data 1.** Original scans for the Western blot analysis in *Figure 5A* (anti-Styxl2, anti-Myh9, anti-Myh10 and anti-Tubulin).

**Source data 2.** A PDF file showing original scans of the relevant Western blot analysis in *Figure 5A* (anti-Styxl2, anti-Myh9, anti-Myh10, and anti-Tubulin) with highlighted bands and sample labels.

**Source data 3.** Original scans for the Western blot analysis in *Figure 5B* (anti-Styxl2, anti-Myh9, anti-Myh10, and anti-Tubulin).

**Source data 4.** A PDF file showing original scans of the relevant Western blot analysis in *Figure 5B* (anti-Styxl2, anti-Myh9, anti-Myh10, and anti-Tubulin) with highlighted bands and sample labels.

**Source data 5.** Original scans for the Western blot analysis in *Figure 5D* (anti-Styxl2, anti-Myh9, anti-Myh10, and Loading).

**Source data 6.** A PDF file showing original scans of the relevant Western blot analysis in *Figure 5D* (anti-Styxl2, anti-Myh9, anti-Myh10, and Loading) with highlighted bands and sample labels.

**Figure supplement 1.** The protein levels of NM IIs in differentiating C2C12 cells and the knockdown efficiency of Myh10-MO.

*Figure 5 continued on next page*

Figure 5 continued

**Figure supplement 1—source data 1.** Original scans for the Western blot analysis in *Figure 5—figure supplement 1A* (anti-MHC, anti-Myh9, anti-Myh10, anti-Styxl2, and anti-Tubulin).

**Figure supplement 1—source data 2.** A PDF file showing original scans of the relevant Western blot analysis in *Figure 5—figure supplement 1A* (anti-MHC, anti-Myh9, anti-Myh10, anti-Styxl2, and anti-Tubulin) with highlighted bands and sample labels.

**Figure supplement 1—source data 3.** Original scans for the Western blot analysis in *Figure 5—figure supplement 1B* (anti-Myh10 and anti-Tubulin).

**Figure supplement 1—source data 4.** A PDF file showing original scans of the relevant Western blot analysis in *Figure 5—figure supplement 1B* (anti-Myh10 and anti-Tubulin) with highlighted bands and sample labels.

## Styxl2 induces the degradation of NM IIs via its C-terminal domain

Our data above showed that loss of Styxl2 caused abnormal accumulation of NM IIs (*Figure 5*), which suggested that Styxl2 normally promote degradation of NM IIs during nascent sarcomere assembly. To test whether this is the case, we co-expressed HA-Myh9 with either Mst1 (as a negative control) or Styxl2 in C2C12 myoblasts. The protein levels of HA-Myh9 were indeed reduced by co-transfected

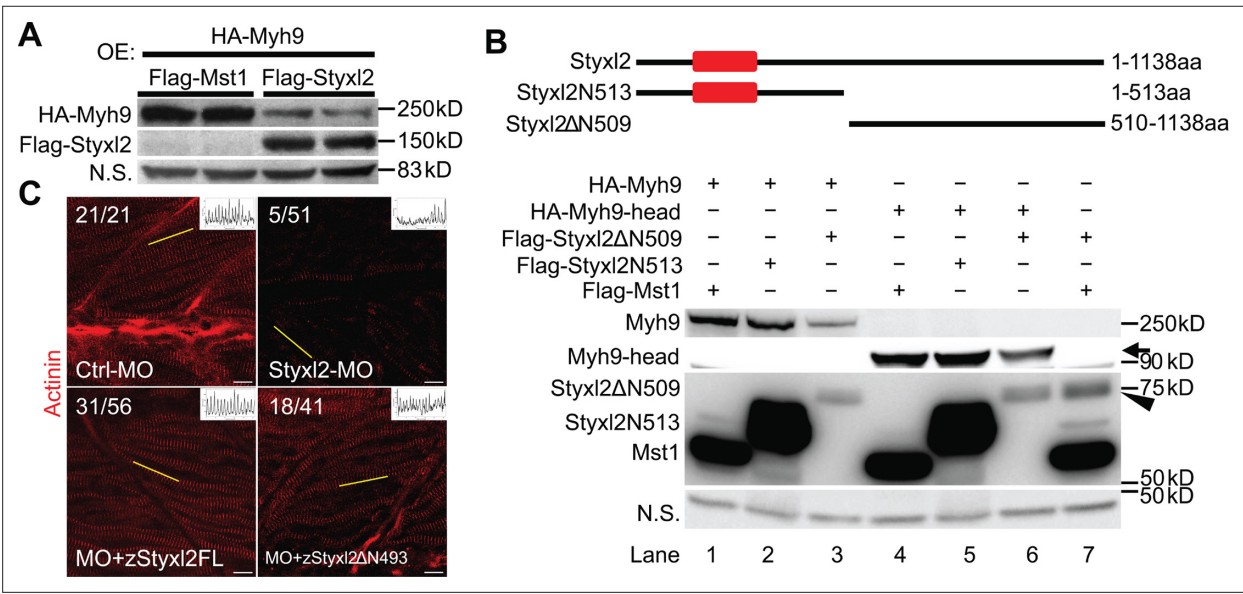

**Figure 6.** The dual-specificity phosphatase catalytic (DSPc) domain of Styxl2 is dispensable for degradation of non-muscle myosin IIs. (**A**) DNA constructs encoding Myh9, Styxl2, or Mst1 (negative control) were co-expressed in C2C12 cells. Soluble whole cell extracts were subjected to Western blot analysis. OE: overexpression. (**B**) The schematic shows the truncated fragments of Styxl2 protein. The red blocks indicate the DSPc domain of Styxl2. HEK 293T cells were transfected with different plasmids as indicated. 24 hr after transfection, cells were harvested, and soluble whole cell extracts were subjected to Western blot analysis. The arrow indicates the head domain of Myh9, and the arrowhead indicates Styxl2 missing the N-terminal 509 aa (Styxl2ΔN509). Mst1: negative control. N.S.: non-specific. (**C**) Zebrafish zygotes were injected with various morpholinos as indicated with or without the co-injected *Styxl2* mRNAs. z*Styxl2*FL: the full-length fish *Styxl2* mRNA; z*Styxl2*ΔN493: the truncated fish *Styxl2* mRNA missing the 5' region encoding the N-terminal 493 aa. Representative α-actinin immunofluorescent images were shown. The number in numerator at the top left corner of each image represents fish embryos showing normal α-actinin staining pattern, while that in denominator represents the total number of embryos examined. The inset at the top right corner of each image shows the actinin intensity along the yellow line. Scale bar: 10 μm.

The online version of this article includes the following source data and figure supplement(s) for figure 6:

**Source data 1.** Original scans for the Western blot analysis in *Figure 6A* (anti-Flag, anti-HA, and N.S.).

**Source data 2.** A PDF file showing original scans of the relevant Western blot analysis in *Figure 6A* (anti-Flag, anti-HA, and N.S.) with highlighted bands and sample labels.

**Source data 3.** Original scans for the Western blot analysis in *Figure 6B* (anti-Flag, anti-HA, and N.S.).

**Source data 4.** A PDF file showing original scans of the relevant Western blot analysis in *Figure 6B* (anti-Flag, anti-HA, and N.S.) with highlighted bands and sample labels.

**Figure supplement 1.** Styxl2 regulates the protein levels of NM IIs without affecting their mRNA.

Styxl2 (*Figure 6A*). We further found that Styxl2-mediated reduction in Myh9 protein levels was not caused by reduced *Myh9* mRNA (*Figure 6—figure supplement 1A*). To determine which part of Styxl2 promotes degradation of NM IIs, we truncated Styxl2 into two fragments (*Figure 6B*, upper panel): an N-terminal fragment containing the DSPc domain (i.e. aa 1–513, also termed Styxl2N513 hereafter) and a C-terminal fragment without the DSPc domain (i.e. aa 510–1138, also named Styxl2ΔN509 hereafter). When we co-transfected 293T cells with a construct encoding HA-Myh9 together with another construct encoding Mst1, Styxl2N513, or Styxl2ΔN509, we found that only Styxl2ΔN509 promoted degradation of Myh9 (*Figure 6B*, lane 3). As we previously showed that the head domains of NM IIs were involved in interacting with Styxl2 (*Figure 4E and F*), we then tested whether the protein levels of the head domain of Myh9 were also regulated by Styxl2. Consistently, only Styxl2ΔN509 promoted degradation of the head domain of Myh9 (*Figure 6B*, lane 6). To further verify this finding in vivo, we microinjected the mRNA encoding the full-length or the C-terminus of zebrafish Styxl2 (zStyxl2), along with the MO for z*Styxl2*, into the zebrafish one-cell embryos. As expected, the defective sarcomere structure caused by *zStyxl2* knockdown was efficiently rescued by overexpressing the mRNA encoding the full-length zStyxl2, as revealed by the α-actinin staining (*Figure 6C*). Importantly, the mRNA encoding the C-terminus of zStyxl2 was nearly as effective as that encoding the full-length zStyxl2 (*Figure 6C*). Similar results were also obtained when we subjected fish to myosin staining (*Figure 6—figure supplement 1B*). Thus, the C-terminus of Styxl2, without the DSPc domain, is both necessary and sufficient to promote degradation of NM IIs during nascent sarcomere assembly.

## Styxl2 promotes the degradation of NM IIs via the autophagy-lysosome pathway

In order to understand how Styxl2 promotes the degradation of NM IIs, we first turned to the ubiquitination/proteasome pathway (*Nandi et al., 2006*). MG132, a specific proteasome inhibitor, was added to C2C12 cells transfected with Styxl2. No abnormal accumulation of NM IIs was found in the presence of MG132 (*Figure 7—figure supplement 1*), suggesting that the proteasome is not involved in Styxl2-mediated degradation of NM IIs. Next, we turned to the autophagy-lysosome-dependent protein degradation system. In C2C12 myoblasts, we co-expressed Myh9 with either Mst1 (control) or Styxl2 together with an siRNA against GFP (control) or ATG5, a key component of the autophagy pathway (*Mizushima et al., 2010*). In the presence of the control siRNA, Styxl2 efficiently promoted degradation of Myh9 as expected (*Figure 7A*, lane 2). However, in the presence of ATG5 siRNA, the protein levels of Myh9 were restored to levels comparable to that of the control even in the presence of the co-expressed Styxl2 (*Figure 7A*, lane 3). This result supported the involvement of the autophagy pathway in Styxl2-mediated degradation of NM IIs. Furthermore, we also examined the degradation of Myh9 induced by Styxl2Δ509 with or without several different autophagy inhibitors (*Figure 7B*). In the absence of any autophagy inhibitors, Styxl2ΔN509 induced degradation of Myh9 as expected (*Figure 7B*, lanes 3 and 4). In the presence of LY294002 (an inhibitor of PI3K), chloroquine (an inhibitor that blocks the binding of autophagosomes to lysosomes), bafilomycin A1 (BafA1, an inhibitor of V-ATPase that can block autophagic flux), and $NH_4Cl$ (an inhibitor of the degradative enzymes inside lysosomes), the protein levels of Myh9 were largely restored to the levels comparable to that in the controls (*Figure 7B*, lanes 1 and 2). Thus, we concluded that the autophagy-lysosome pathway is involved in the degradation of NMIIs induced by Styxl2.

It is known that some ubiquitinated proteins (such as sarcomeric titin) are degraded by selective autophagy (*Bogomolovas et al., 2021*; *Chen et al., 2019*; *Kirkin et al., 2009*; *Pankiv et al., 2007*; *Shaid et al., 2013*; *Zaffagnini and Martens, 2016*). A recent report showed that Myh9 could also undergo Nek9-mediated selective autophagy (*Yamamoto et al., 2021*), suggesting that Myh9 is ubiquitinated. To find out whether Styxl2 promotes ubiquitination of NM IIs, we co-transfected HEK 293T cells with plasmids encoding Myh9 and ubiquitin together with a plasmid encoding either Mst1 or Styxl2. Compared to the Mst1 control, we found that Styxl2 indeed promoted the ubiquitination of Myh9 (*Figure 7C*).

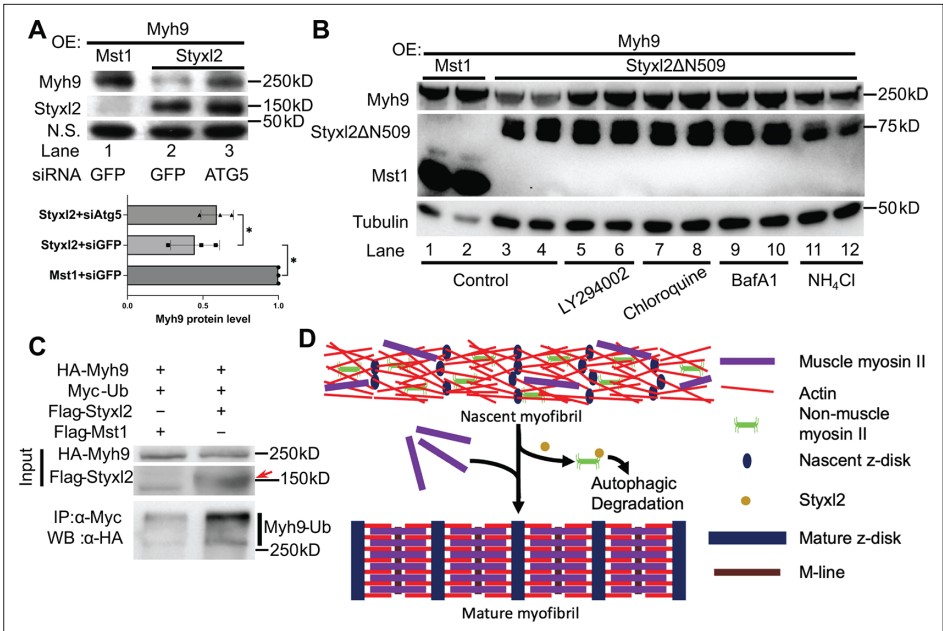

**Figure 7.** The Styxl2-mediated degradation of non-muscle myosin IIs is autophagy dependent. (**A**) Plasmids encoding Myh9, Styxl2, or Mst1 (negative control) were co-expressed in C2C12 cells together with various siRNAs as indicated. Soluble whole cell extracts were subjected to Western blot analysis. OE: overexpression. N.S.: non-specific. The protein level of Myh9 was quantified at the bottom panel. *p-value <0.05. (**B**) HEK 293T cells were transfected with various constructs as indicated. Various inhibitors of the autophagy-lysosome pathway (50 µM LY294002, 100 µM Chloroquine, 100 nM BafA1, 20 mM NH₄Cl) were added to the cell culture 6 hr before harvest. OE: overexpression. (**C**) HEK 293T cells were co-transfected with the plasmids as indicated. 18 hr later, 100 nM of BafA1 was added to the culture medium. After another 6 hr, the soluble whole cell lysates were subjected to immunoprecipitation (IP) with an anti-Myc antibody, and the immunoprecipitated proteins were further analysed by Western blot (WB). The red arrow indicates Flag-Styxl2. (**D**) The schematic shows that Styxl2 promotes sarcomere assembly by binding to and targeting non-muscle myosin IIs for degradation, which facilitates their eventual replacement by muscle myosin II during sarcomere maturation.

The online version of this article includes the following source data and figure supplement(s) for figure 7:

**Source data 1.** Original scans for the Western blot analysis in *Figure 7A* (anti-Styxl2, anti-Myh9, and N.S.).

**Source data 2.** A PDF file showing original scans of the relevant Western blot analysis in *Figure 7A* (anti-Styxl2, anti-Myh9, and N.S.) with highlighted bands and sample labels.

**Source data 3.** Original scans for the Western blot analysis in *Figure 7B* (anti-Myh9, anti-Styxl2ΔN509, anti-Mst1, and anti-Tubulin).

**Source data 4.** A PDF file showing original scans of the relevant Western blot analysis in *Figure 7B* (anti-Myh9, anti-Styxl2ΔN509, anti-Mst1, and anti-Tubulin) with highlighted bands and sample labels.

**Source data 5.** Original scans for the Western blot analysis of input and IP in *Figure 7C* (anti-Flag and anti-HA).

**Source data 6.** A PDF file showing original scans of the relevant Western blot analysis in *Figure 7C* (anti-Flag and anti-HA) with highlighted bands and sample labels.

**Figure supplement 1.** Proteasome is not involved in Styxl2-mediated degradation of NM IIs.

**Figure supplement 1—source data 1.** Original scans for the Western blot analysis in *Figure 7—figure supplement 1* (anti-Myh9, anti-Myh10, anti-Styxl2, and Loading).

**Figure supplement 1—source data 2.** A PDF file showing original scans of the relevant Western blot analysis in *Figure 7—figure supplement 1* (anti-Myh9, anti-Myh10, anti-Styxl2, and Loading) with highlighted bands and sample labels.

## Discussion
### Styxl2 is a pseudophosphatase that functions independently of its conserved phosphatase domain
Pseudophosphatases are present in multicellular organisms ranging from *C. elegans* to humans

(*Hinton, 2019*; *Reiterer et al., 2020*). Moreover, pseudophosphatases have also been identified in plants (*Bellec et al., 2002*). Recently, 26 pseudophosphatase domains, including the phosphatase domain of Styxl2, were bioinformatically identified in the human genome (*Chen et al., 2017*). Due to the absence of one or several conserved amino acids including the catalytic cysteine, pseudophosphatases are not expected to possess the phosphatase activity. STYX, the first molecularly characterized pseudophosphatase, was identified in 1995 as a phosphoserine/threonine/tyrosine-binding protein (*Wishart et al., 1995*). The substitution of the catalytic cysteine by glycine in the active site motif leads to the loss of its catalytic activity (*Reiterer et al., 2013*; *Wishart et al., 1995*). In addition to STYX, the functions of several other pseudophosphatases from *C. elegans* to humans have also been characterized (*Cheng et al., 2009*; *Hinton et al., 2010*; *Maruyama et al., 2007*; *Nakhro et al., 2013*; *Tomar et al., 2019*). These reports unequivocally demonstrate that pseudophosphatases are not simply evolution relics but play essential roles in multicellular organisms.

Pseudophosphatases exert their functions via diverse modes including direct competition with other proteins by binding to a phospho-residue-containing motif, modulation of the phosphatase activity by binding to an active phosphatase, sequestration of target proteins in specific subcellular locations, and integration of signals by simultaneous binding with multiple proteins (*Hinton, 2019*; *Reiterer et al., 2020*). The original report on STYX showed that mutation of the glycine in the conserved H$\underline{G}$X$_5$R motif back to cysteine restored the phosphatase activity (*Wishart et al., 1995*), suggesting that the three-dimensional structure of the phosphatase domain in STYX is still folded properly. Thus, some pseudophosphatases are thought to function by binding to specific phosphorylated substrates, which is the case for pseudophosphatases EGG-4/5 and Pas2 (*Bellec et al., 2002*; *Cheng et al., 2009*; *Da Costa et al., 2006*). In these examples, pseudophosphatases bind the phosphorylated sites of selected target proteins through their inactive phosphatase domains and prevent them from binding other interacting proteins or active phosphatases. However, this is not the only way pseudophosphatases function. MTMR13, a pseudophosphatase, functions by binding to MTMR2, an active lipid phosphatase, to regulate its phosphatase activity. Mutations in either *MTMR13* or *MTMR2* are causally linked to human Type 4B Charcot-Marie-Tooth disease, an inherited disorder of the peripheral nervous system with abnormal nerve myelination (*Berger et al., 2006*; *Robinson and Dixon, 2005*).

Unlike several other characterized pseudophosphatases that function via their conserved phosphatase domains, Styxl2 appears to function independently of its phosphatase domain as the phosphatase domain of Styxl2 is dispensable for its ability to induce degradation of NM IIs and to rescue the sarcomere defects in the *Styxl2* morphant zebrafish (*Figure 6B and C*). Such a phosphatase domain-independent mode of action is also observed in His Domain Protein Tyrosine Phosphatase (HD-PTP or PTPN23), a pseudophosphatase known to facilitate endosomal cargo sorting and the degradation of multiple membrane receptors including epidermal growth factor receptor, platelet-derived growth factor receptor, MHC class I and Integrin α5β1 (*Doyotte et al., 2008*; *Kharitidi et al., 2015*; *Ma et al., 2015*; *Parkinson et al., 2015*). Interestingly, a truncated fragment of HD-PTP without its phosphatase domain is still capable of promoting endosomal trafficking of cargo proteins like epidermal growth factor (EGF) (*Doyotte et al., 2008*).

## Styxl2 facilitates de novo sarcomere assembly by promoting autophagy-dependent degradation of non-muscle myosin IIs

Although absent in invertebrates, Styxl2 is expressed and conserved in vertebrates. The specific involvement of Styxl2 in sarcomere assembly is supported by its restricted expression in striated muscles. Interestingly, mutation in the coding region of *STYXL2* and abnormal CpG methylation at the *STYXL2* locus were detected in some patients with congenital heart defects (CHD), suggesting a possible link of *STYXL2* to CHD (*Arrington et al., 2012*; *Radhakrishna et al., 2016*). Moreover, its role in sarcomere assembly is also conserved in vertebrates as interference of the gene expression in zebrafish by either a transposon integration or morpholino oligos leads to severe defects in sarcomere assembly (*Fero et al., 2014*; *Figure 1D and E*). However, Styxl2 is not essential for sarcomere assembly as sarcomeres in *Drosophila* flight muscles can normally assemble in the absence of a Styxl2 homolog. Furthermore, we could still find residual sarcomeres in skeletal muscles of conditional *Styxl2* KO mice (driven by either *Pax7$^{Cre}$* or *Myf5$^{Cre}$*) (*Figure 2—figure supplement 1E*).

Although the highly-ordered sarcomere structure has been known for many years, it remains controversial how sarcomeres correctly assemble. Several different but not mutually exclusive models have been proposed (*Rui et al., 2010*; *Sanger et al., 2006*; *Sanger et al., 2017*). Studies in *Drosophila*, *C. elegans*, and mice underscore an essential role for integrin in initiating the assembly of sarcomeres (*Rui et al., 2010*; *Sparrow and Schöck, 2009*). For example, β1 integrin was shown to be essential for sarcomere assembly in mice (*Schwander et al., 2003*). Among the existing sarcomere assembly models, the prevailing model of sarcomere assembly is called premyofibril model (*Rhee et al., 1994*; *Sanger et al., 2006*; *Sanger et al., 2017*). In this model, α-actinin, skeletal muscle actin, and non-muscle myosin IIs (NM IIs) first assemble to form mini-sarcomeres or premyofibrils at the cell periphery. Then, additional sarcomeric proteins like titin, nebulin, troponin, and myomesin will be incorporated and NM IIs will be eventually replaced by skeletal muscle myosin in mature sarcomeres (*White et al., 2014*; *White et al., 2018*). Consistent with this model, germline knockout of NM IIB (i.e. Myh10) in mice led to early lethality between E14.5 and birth and sarcomere structural defects in embryonic cardiomyocytes (*Tullio et al., 1997*). Similarly, defective sarcomeres were also observed in embryonic muscles of stage-17 mutant fly null for *zipper* (*zip*), the *Drosophila* gene encoding the heavy chain of NM II (*Bloor and Kiehart, 2001*; *Loison et al., 2018*). Moreover, a recent paper using human iPSC-derived cardiomyocytes demonstrated that nascent sarcomeres assemble directly from muscle actin stress fibers (*Fenix et al., 2018*). Both MYH9 and MYH10 are present in muscle actin stress fibers with MYH10 being essential for sarcomere assembly. Consistent with the premyofibril model, NM IIs are found to be replaced by muscle myosin IIs in mature sarcomeres (*Fenix et al., 2018*). Nevertheless, the view of the involvement of NM IIs in early sarcomere assembly is also challenged by several studies. Unlike germline knockout of NM IIB, cardiac-specific knockout of NM IIB in adult mice did not generate obvious defects in sarcomere structures (*Ma et al., 2009*). Although NM IIA is not detected in embryonic hearts, loss of cardiac NM IIB led to increased expression of NM IIA, suggesting that NM IIA may compensate for the loss of NM IIB in NM IIB-null embryonic cardiomyocytes (*Fenix et al., 2018*; *Tullio et al., 1997*). Moreover, it was also suspected that the cardiomyocyte-specific deletion of NM IIB occurs after the initial sarcomere assembly due to the expression timing of the Cre recombinase under the control of the cardiac α-MHC promoter (*Fenix et al., 2018*; *Ma et al., 2009*). Furthermore, a recent study using CRISPR/Cas9 to knock out *MYH9* and *MYH10* genes either individually or in combination in human iPSC-derived cardiomyocytes showed that neither gene is required for sarcomere assembly in their cell culture models (*Chopra et al., 2018*). It is puzzling at present why the results on the involvement of NM IIs in de novo sarcomere assembly differ in the two studies by Fenix et al. and Chopra et al. even though both studies used human iPSC-derived cardiomyocytes to assay de novo sarcomere assembly in cell culture.

It also remains unclear how NM IIs are replaced by muscle myosin during de novo sarcomere assembly in vertebrates. Our current study showed that Styxl2 interacts with NM IIs and induces their degradation via the autophagy-lysosome pathway. Autophagy is known to regulate sarcomere integrity (*Masiero et al., 2009*; *Nakai et al., 2007*). In particular, selective autophagy is known to degrade multiple ubiquitinated sarcomeric proteins including titin and filamin (*Arndt et al., 2010*; *Bogomolovas et al., 2021*). p62/SQSTM1 and NBR1 have been shown to act as adaptors linking ubiquitinated substrates to Atg8/LC3 that is localized on the membrane of autophagosome (*Kirkin et al., 2009*; *Pankiv et al., 2007*). Interestingly, a recent study on ciliogenesis showed that MYH9 is also degraded by selective autophagy in an NEK9-mediated manner (*Yamamoto et al., 2021*). Consistently, we found that Styxl2 promotes the ubiquitination of Myh9 (*Figure 7C*). It remains to be established whether Nek9 is involved in Styxl2-mediated autophagic degradation of NM IIs and whether ubiquitination of NM IIs is dependent on Nek9-mediated phosphorylation during myofibrillogenesis.

Our data showed that Styxl2 plays an indispensable role during embryonic myogenesis as conditional deletion of *Styxl2* during mouse embryo development using either *Pax7^Cre* or *Myf5^Cre* resulted in severe disruption of sarcomere structures in skeletal muscles and perinatal lethality (*Figure 2* and *Figure 2—figure supplement 1*). In contrast, inducible deletion of *Styxl2* in adult muscles only led to mild defects in sarcomere structures (*Figure 3F–J*). It is likely that additional unknown molecules can compensate for the loss of Styxl2 in adult regenerating muscles. Such distinct roles of Styxl2 during development are also supported by differential expression patterns of Styxl2 and NM IIs: both are abundantly expressed in new-born and juvenile mice. While the protein levels of Myh9 and Myh10 start to drop one week after birth and become undetectable by three weeks after birth, that of Styxl2

is relatively stable in the first month after birth but starts to drop sharply by two months after birth (*Figure 5A*). This is also consistent with our findings that Styxl2 is not required for the maintenance of the sarcomeres in adult muscles (*Figure 3A–D*).

In summary, our current study shows that Styxl2 critically regulates de novo sarcomere assembly by binding to NM IIs, enhancing their ubiquitination, and promoting their autophagic degradation, which ultimately facilitates sarcomere maturation in vivo (*Figure 7D*).

# Materials and methods

## Key resources table

| Reagent type (species) or resource | Designation | Source or reference | Identifiers | Additional information |
|---|---|---|---|---|
| Gene (*M. musculus*) | Styxl2 | GenBank | MGI:MGI:2685055 | |
| Gene (*D. rerio*) | zStyxl2 (dusp27) | GenBank | ZFIN:ZDB-GENE-140513–1 | |
| Genetic reagent (*M. musculus*) | Styxl2flox/flox | This paper | | See Materials and methods, Section 1. |
| Genetic reagent (*M. musculus*) | EIIA-Cre | The Jackson Laboratory | Strain #:003724 | |
| Genetic reagent (*M. musculus*) | Pax7CreERT2/+ | The Jackson Laboratory | Strain #:017763 | |
| Genetic reagent (*M. musculus*) | Myf5Cre/+ | *Tallquist et al., 2000* | | |
| Genetic reagent (*M. musculus*) | Pax7Cre/+ | *Keller et al., 2004* | | |
| Genetic reagent (*M. musculus*) | Tg (HSA-MerCreMer) | *McCarthy et al., 2012* | | |
| Strain, strain background (*D. rerio*) | ABSR wild type fish | Prof. Zilong Wen (HKUST) | | |
| Cell line (*M. musculus*) | C2C12 | ATCC | CRL-1772 | |
| Cell line (*Homo-sapiens*) | HEK 293T | ATCC | CRL-3216 | |
| Transfected construct (*M. musculus*) | Flag-Styxl2 | This paper | | See Materials and methods, Section 2. |
| Transfected construct (*M. musculus*) | Flag-Styxl2ΔN509 | This paper | | See Materials and methods, Section 2. |
| Transfected construct (*M. musculus*) | Flag-Styxl2N513 | This paper | | See Materials and methods, Section 2. |
| Transfected construct (*M. musculus*) | Myosin-IIA-GFP | addgenen | #38297 | |
| Transfected construct (*M. musculus*) | HA-Myh9 | This paper | | See Materials and methods, Section 2. |
| Transfected construct (*M. musculus*) | HA-Myh9-head | This paper | | See Materials and methods, Section 2. |
| Transfected construct (*M. musculus*) | HA-Myh9-tail | This paper | | See Materials and methods, Section 2. |
| Transfected construct (*M. musculus*) | HA-Myh10-head | This paper | | See Materials and methods, Section 2. |
| Transfected construct (*M. musculus*) | HA-Myh10-tail | This paper | | See Materials and methods, Section 2. |
| Transfected construct (*E. coli*) | pcDNA3.1 mycBioID | addgene | #35700 | Re-cloned into pcDNA3.0. |
| Transfected construct (*D. rerio*) | zStyxl2 | This paper | | See Materials and methods, Section 2. |
| Transfected construct (*D. rerio*) | zStyxl2ΔN493 | This paper | | See Materials and methods, Section 2. |
| Antibody | anti-Jak1 (Rabbit Polyclonal) | Upstate | Cat#: 06–272 | IB(1:2000) |

*Continued on next page*

*Continued*

| Reagent type (species) or resource | Designation | Source or reference | Identifiers | Additional information |
|---|---|---|---|---|
| Antibody | anti-Stat1 (Rabbit Polyclonal) | Upstate | Cat#: 06–501 | IB(1:2000) |
| Antibody | anti-β-Tubulin (Mouse monoclonal) | Sigma | Cat#: T4026 | IB(1:5000) |
| Antibody | anti-sarcomere MHC (Mouse monoclonal) | DSHB | MF20 | IB(1:1000) IF(1:200) |
| Antibody | anti-GAPDH (Mouse Monoclonal) | Ambion | Cat#: AM4300 | IB(1:10000) |
| Antibody | anti-Myh9 (Rabbit Polyclonal) | Cell Signaling | Cat#: 3403 | IB(1:2000) IF(1:200) |
| Antibody | anti-Myh10 (Mouse monoclonal) | Santa Cruz | Cat#: sc-376942 | IB(1:1000) |
| Antibody | anti-Flag (Mouse monoclonal) | Sigma | F3165 | IF(1:5000) |
| Antibody | anti-HA (Mouse monoclonal) | Roche | 12CA5 | IB(1:5000) |
| Antibody | anti-Styxl2 (Rabbit Polyclonal) | This paper | | See Materials and methods, Section 7. IB(1:500) |
| Antibody | anti-MEF2A (Rabbit Polyclonal) | This paper | | Immunize rabbits with mouse MEF2A. |
| Antibody | anti-Flag (Rabbit Polyclonal) | Sigma | F7425 | IF(1:200) |
| Antibody | anti-HA (Rabbit polyclonal) | Santa Cruz | Cat#: sc-805 | IF(1:200) |
| Antibody | anti-Myc (Mouse monoclonal) | Santa Cruz | Cat#: sc-40 | IB(1:1000) |
| Antibody | anti-α-Actinin (Mouse monoclonal) | Sigma | Cat#: A7732 | IF(1:200) |
| Antibody | anti-Prox1 (Rabbit polyclonal) | Angiobio | Cat#: 11–002 P | IF(1:200) |
| Antibody | anti-slow myosin heavy chain (Mouse monoclonal) | DSHB | F59 | IF(1:200) |
| Antibody | anti-myosin heavy chain (Mouse monoclonal) | DSHB | A4.1025 | IF(1:200) |
| Recombinant DNA reagent | pRetroX-Tet-On Advanced (plasmid) | Dr. Yusong Guo (HKUST) | | |
| Sequence-based reagent | Styxl2-F | This paper | PCR primers | GCCCATCCACCTCTCCTC |
| Sequence-based reagent | Styxl2-R | This paper | PCR primers | GCTCCTGTCATCCATCTTCTC |
| Sequence-based reagent | Myh9-F | This paper | PCR primers | GGCCCTGCTAGATGAGGAGT |
| Sequence-based reagent | Myh9-R | This paper | PCR primers | CTTGGGCTTCTGGAACTTGG |
| Sequence-based reagent | Myh10-F | This paper | PCR primers | GGAATCCTTTGGAAATGCGAAGA |
| Sequence-based reagent | Myh10-R | This paper | PCR primers | GCCCCAACAATATAGCCAGTTAC |
| Sequence-based reagent | Gapdh-F | This paper | PCR primers | CCCACTCTTCCACCTTCG |
| Sequence-based reagent | Gapdh-R | This paper | PCR primers | TCCTTGGAGGCCATGTAG |
| Sequence-based reagent | siGFP (negative control) | This paper | siRNA | GCTGACCCTGAAGTTCATC |
| Sequence-based reagent | siJak1 | This paper | siRNA | GCCTGAGAGTGGAGGTAAC |
| Sequence-based reagent | siStat1 | This paper | siRNA | GGATCAAGTCATGTGCATA |
| Sequence-based reagent | siATG5 | This paper | siRNA | GGCTCCTGGATTATGTCAT |
| Sequence-based reagent | Ctrl (negative control) | This paper | morpholino | AGCACACAAAGGCGAAGGTCAACAT |
| Sequence-based reagent | Styxl2 | This paper | morpholino | GCTGATCCTCCACAGACGACGCCAT |
| Sequence-based reagent | Myh10 | This paper | morpholino | CTTCACAAATGTGGTCTTACCTTGA |
| Sequence-based reagent | Atg5 | This paper | morpholino | CACATCCTTGTCATCTGCCATTATC |
| Chemical compound, drug | Cardiotoxin | Sigma Aldrich | Cat#: 217503 | |
| Software, algorithm | ImageJ | National Institutes of Health | | |

## Animals

ABSR wild-type zebrafish were maintained following standard procedures and the zygotes were collected for microinjection experiments. *Pax7*^CreERT2/+^ mice (Stock No: 017763) were purchased from the Jackson Laboratory (Bar Harbor, ME, USA). *Myf5*^Cre/+^, *Pax7*^Cre/+^, and Tg (HSA-MerCreMer) mice were generous gifts from Drs. Michael Rudnicki (Ottawa Hospital Research Institute, ON, Canada), Mario Capecchi (University of Utah, UT, USA), and Karyn Esser (University of Florida, FL, USA), respectively. *Styxl2*^flox/flox^ mice were generated by the Model Animal Research Centre (Nanjing University, Nanjing, China). To delete *Styxl2* in adult mice, tamoxifen dissolved in corn oil (15 mg/ml) was injected into adult mice intraperitoneally at 5 µl per gram of body weight for five consecutive days followed by another five doses of tamoxifen injection every 3–4 days. All animals were maintained and handled following the protocols approved by the Animal Ethics Committee of the Hong Kong University of Science and Technology.

## Cell culture and DNA constructs

C2C12 cells and HEK 293T cells were purchased from ATCC (Manassas, VA, USA). To ensure the cell quality, low-passage cells were used and the cell morphology was regularly examined. For C2C12 cells, the differentiation potential was also regularly examined by the expression of selected differentiation markers. In addition, DAPI staining was employed to detect the mycoplasma contamination in cell culture. C2C12 cells were maintained in growth medium (GM) (Dulbecco's modified Eagle's medium supplemented with 20% fetal bovine serum, 100 Units/ml penicillin, and 100 µg/ml streptomycin) and differentiation was induced with differentiation medium (DM) (Dulbecco's modified Eagle's medium supplemented with 2% horse serum, 100 Units/ml penicillin, and 100 µg/ml streptomycin). HEK 293T cells were maintained in Dulbecco's modified Eagle's medium supplemented with 10% fetal bovine serum, 100 Units/ml penicillin, and 100 µg/ml streptomycin. Mouse Styxl2, Styxl2ΔN509, and Styxl2N513 were amplified from a mouse muscle cDNA library and inserted into a modified pcDNA3.0 vector containing a Flag-expressing cassette to generate constructs encoding Flag-Styxl2, Flag-Styxl2ΔN509, and Flag-Styxl2N513. BirA* was amplified from pcDNA3.1 mycBioID (addgene, #35700) and an HA tag was added to its C-terminus. The cDNA encoding BirA*-HA was fused in-frame with that encoding mouse Styxl2 and cloned into the pRetroX-Tight-Pur vector (kindly provided by Dr. Yusong Guo at HKUST) to get pTight-Styxl2-BirA*-HA. Mouse Myh9, Myh9-head, and Myh9-tail were amplified from a plasmid encoding Myosin-IIA-GFP (addgene, #38297) and inserted into a modified pcDNA3.0 vector containing an HA-expressing cassette to generate constructs encoding HA-Myh9, HA-Myh9-head, and HA-Myh9-tail. Mouse Myh10-head and Myh10-tail were amplified from a mouse C2C12 cell cDNA library and inserted into a modified pcDNA3.0 vector containing an HA-expressing cassette to generate constructs encoding HA-Myh10-head and HA-Myh10-tail. Fish Styxl2 and Styxl2ΔN493 were amplified from a zebrafish cDNA library and inserted into pCS2+ vector to generate pCS2+-zStyxl2 and pCS2+-zStyxl2ΔN493.

## RNA extraction and RT-qPCR

Total RNA was extracted from C2C12 cells using the TRIzol Reagent (Invitrogen) following the manufacturer's instructions. The reverse transcription was performed using an oligo-dT primer and the ImRrom-II Reverse Transcription System (Promega). The ABI 7700 Sequence Detection System (PE Applied-Biosystems) was used for qPCR and the relative fold change of genes of interest was analysed based on the $2^{-\triangle\triangle Ct}$ method (***Livak and Schmittgen, 2001***). The qPCR primers used were listed below:

> *Styxl2*-F: 5'- GCCCATCCACCTCTCCTC-3'
> *Styxl2*-R: 5'-GCTCCTGTCATCCATCTTCTC-3'
> *Myh9*-F: 5'-GGCCCTGCTAGATGAGGAGT-3'
> *Myh9*-R: 5'-CTTGGGCTTCTGGAACTTGG-3'
> *Myh10*-F: 5'-GGAATCCTTTGGAAATGCGAAGA-3'
> *Myh10*-R: 5'-GCCCCAACAATATAGCCAGTTAC-3'
> *Gapdh*-F: 5'-CCCACTCTTCCACCTTCG-3'
> *Gapdh*-R: 5'-TCCTTGGAGGCCATGTAG-3'

## siRNA transfection

The following siRNAs were used to knock down specific genes. They were transfected into C2C12 cells using the Lipofectamine RNAiMax reagent (Thermo Fisher Scientific) following the manufacturer's instructions.

> *GFP* (negative control): 5'-GCTGACCCTGAAGTTCATC-3'
> *Jak1*: 5'-GCCTGAGAGTGGAGGTAAC-3'
> *Stat1*: 5'-GGATCAAGTCATGTGCATA-3'
> *Atg5*: 5'-GGCTCCTGGATTATGTCAT-3'

## Microarray

Total mRNA were extracted from duplicated samples and subjected to microarray analysis using the Affymetrix GeneChip Mouse Genome 430 2.0 Array. The data were analysed by the GeneSpring software and deposited to GEO (Accession number: GSE262242).

## Plasmid transfection

The Lipofectamine Transfection Reagent (Thermo Fisher Scientific) and the PLUS Reagent (Thermo Fisher Scientific) were used to transfect plasmids into C2C12 myoblasts following the manufacturer's instructions. Polyethylenimine (PEI, Polysciences) was used to transfect plasmids into HEK 293T cells.

## Western blot

For cultured cells, Western blot was performed according to the standard procedures. For mouse tissues, the samples were homogenized using a T-25 digital IKA ULTRA-TURRAX disperser in a cell lysis buffer (25 mM Tris-HCl pH7.4, 150 mM NaCl, 1 mM EDTA, 1% NP-40, 5% glycerol) with a cocktail of protease inhibitors (20 mM *p*-nitrophenylphosphate, 20 mM β-glycerolphosphate, 0.5 mM phenylmethylsulfonyl fluoride, 2 μg/ml aprotinin, 0.5 μg/ml leupeptin, 0.7 μg/ml pepstatin, 50 μM sodium vanadate). The soluble lysates were used for Western blot. The primary antibodies used include anti-Jak1 (Upstate, 06–272), anti-Stat1 (Upstate, 06–501), anti-β-Tubulin (Sigma, T4026), anti-sarcomere myosin heavy chain (MHC) (DSHB, MF20), anti-GAPDH (Ambion, AM4300), anti-Myh9 (Cell Signaling, 3403), anti-Myh10 (Santa Cruz, sc-376942), anti-Flag (Sigma, F3165), and anti-HA (Roche, 12CA5). The Styxl2 antiserum was generated in-house by injecting rabbits with recombinant proteins spanning aa 2–142 of Styxl2 fused in-frame with glutathione S-transferase. The antiserum was purified with protein-G beads. The streptavidin-HRP was purchased from GE Healthcare (#RPN1231).

## Immunostaining

C2C12 myoblasts were fixed with 4% paraformaldehyde (PFA) for 10 min at room temperature, and then washed with 0.1% PBST. Then samples were permeabilized with 0.5% PBST for 20 min. Zebrafishes were fixed with PFA for 2–4 hr at room temperature and washed with 0.1% PBST. Then the fishes were permeabilized with cold acetone for 8 min at –20 °C. The following procedures are the same for two types of samples. After permeabilization, the samples were washed with 0.1% PBST and blocked with 4% IgG-free BSA for 1 hr and then incubated with the primary antibody at 4 °C overnight. Then samples were washed with 0.1% PBST and incubated with the secondary antibody at room temperature for 1 hr. The DNA was stained with 100 ng/ml 4',6-diamidino-2-phenylindole (DAPI). For fish slow muscles, the nucleus was stained with an anti-Prox1 antibody. The primary antibodies used include anti-Flag (Sigma, F7425), anti-HA (Santa Cruz, sc-805), anti-α-Actinin (Sigma, A7732), anti-Prox1 (Angiobio, 11–002 P), anti-slow myosin heavy chain (DSHB, F59), anti-myosin heavy chain (DSHB, A4.1025), and anti-sarcomere myosin heavy chain (MHC) (DSHB, MF20).

## Microinjection of zebrafish zygotes

The zygotes of zebrafish were collected in the morning. 2 nl of morpholino-containing solution was injected into fish zygotes as reported before (*Jin et al., 2009*). The concentration of morpholinos used for microinjection was 0.3~0.5 mM. A control morpholino targeting a non-related gene was used as negative control (Ctrl-MO). Morpholinos used were synthesized by Gene Tools (USA) and their sequences were listed below:

Ctrl-MO: 5'-AGCACACAAAGGCGAAGGTCAACAT-3'
*Styxl2*-MO: 5'-GCTGATCCTCCACAGACGACGCCAT-3'
*Myh10*-MO: 5'-CTTCACAAATGTGGTCTTACCTTGA-3'
*Atg5*-MO:5'-CACATCCTTGTCATCTGCCATTATC-3'

The DNA constructs of pCS2+-zStyxl2 (with several synonymous mutations near the start codon of zStyxl2 to make it MO-resistant) and pCS2+-zStyxl2ΔN493 were linearized and used as templates for in vitro transcription. The mRNA was synthesized using the mMESSAGE mMACHINE SP6 Kit (Ambion, AM1340) followed by purification with the MEGAclear Transcription Clean-Up Kit (Invitrogen, AM1908). The purified mRNA was dissolved in RNase-free water and diluted to 150 ng/μl immediately before microinjection.

## Transmission electron microscopy

The heart and skeletal muscles were processed following the published protocols (*Graham and Orenstein, 2007*). Then the samples were examined by a Hitachi H7650 transmission electron microscope (Hitachi High-Technologies). The sarcomere length was measured from electron microscopic images with a magnification of 50,000 X. The length of 3–5 sarcomeres was measured for each mouse and a total of six mice were examined. The results were presented as mean ± SD. The unpaired Student's t-test was performed and $p<0.05$ was considered statistically significant.

## Measurement of muscle force generation in vitro

The gastrocnemius muscle of left hindlimb was carefully dissected from mice together with femur condyle and Achilles tendon. The twitch and tetanic force generated by the dissected gastrocnemius muscle and the muscle cross-sectional area (MCSA) were measured following published protocols (*Guo et al., 2016*). The force generated was recorded by the Dynamic Muscle Control system (DMC v5.4; Aurora Scientific) and analysed by the Dynamic Muscle Analysis system (DMA v3.2; Aurora Scientific). Both twitch and tetanic force were then normalized to the MCSA.

## BioID and mass spectrometry

C2C12 myoblasts were infected by a retrovirus carrying pRetroX-Tet-On Advanced (kindly provided by Dr. Yusong Guo, the Hong Kong University of Science and Technology) and selected by 1 mg/ml G418 to get the Tet-on cells. Then the Tet-on cells were infected by a retrovirus carrying pTight-Styxl2-BirA*-HA and selected by 1.5 μg/ml puromycin to obtain stable Tet-Styxl2-BirA*-HA cells. Thus, the expression of Styxl2-BirA*-HA is controlled by the Tet-on system in these stable C2C12 myoblasts. These stable cells were seeded in GM and cultured for 24 hr before being induced to differentiate in DM for 48 hr. The medium was then replaced with DM containing 50 μM biotin (control) or 50 μM biotin plus 1 μg/ml doxycycline. After another 24 hr, the control and doxycycline-treated cells were processed for mass spectrometry analysis following the published protocols (*Roux et al., 2018*). Briefly, the biotinylated proteins were enriched by Dynabeads MyOne Streptavidin C1 (Invitrogen, #65001). The proteins bound to the beads were reduced with 10 mM Tris(2-carboxyethyl)phosphine hydrochloride (Sigma, C4706) at 40 °C for 30 min and alkylated with 20 mM Iodoacetamide (Sigma, I1149) in the dark for 30 min to prevent re-formation of the disulfide bond. The samples were digested with Trypsin Gold (Promega, V5280) overnight at 37 °C on a Thermomixer at 700 rpm followed by centrifugation. The supernatant was transferred to a new tube, acidified to pH2-3 by trifluoroacetic acid (Thermo Fisher, 85183), and desalted with a Pierce C18 spin column (Thermo Fisher Scientific, 89873). The tryptic peptides were labelled with iTRAQ reagents (AB Sciex, 4390812) following the manufacturer's instructions. Briefly, peptides from control cells were labelled with iTRAQ tag 114 and those from doxycycline-treated cells were labelled with tag 115. The labelled samples were mixed and fractionated by an UltiMate 3000 RSLCnano system (Thermo Fisher Scientific). A C18 Acclaim PepMap RSLC analytical column (75 μm×250 mm, 2 μm, 100 Å) with a C18 nano Viper trap-column (0.3 mm × 5 mm, Thermo Fisher Scientific) was used for peptide elution and separation at a flow rate of 300 nl/min. The mobile phase contained buffer A (0.1% formic acid, 2% acetonitrile) and B (0.1% formic acid, 98% acetonitrile). The gradient was set as follows: 0 min, 3% B; 10 min, 3% B; 12 min, 7% B; 62 min, 20% B; 64 min, 30% B; 65 min, 80% B; 73 min, 80% B; 73.1 min, 3% B; and 90 min, 3% B. MS data were then acquired using an Orbitrap Fusion Lumos mass spectrometer (Thermo Fisher Scientific) in a positive mode, with the following settings: (1) MS1 scan: 400–1600 m/z; resolution:

120,000; automatic gain control (AGC): 400,000; and maximum injection time: 50 ms; (2) The collision energy was set at 35% and orbitrap was used for MS2 scan as well; (3) MS2 scan starts from 100 m/z; resolution: 30,000; AGC: 50,000; and maximum injection time: 250 ms; (4) Exclusion window was set for 40 s; (5) The intensity threshold was set at 25,000. Data analysis was carried out using Mascot Deamon version 2.5.0 (Matrix Science) with standard settings for each instrument type and the data were searched against the *Mus musculus* database downloaded from SwissProt. A peptide tolerance of 10 ppm and fragment ion tolerance of 20 ppm were used. Carbamidomethylation of cysteine was specified as fixed modification, while oxidation of methionine and acetylation of protein N termini were set as variable modifications. The false discovery rate was set at 0.01 for proteins and peptide spectrum match. The 115/114 iTRAQ ratio of one protein was calculated based on the peak intensity of reporter ions by the software, and defined as Enrichment score for proteins in doxycycline-treated cells over that in control cells. The *p* values for each protein generated by the software were used for gene ontology analysis.

## Gene ontology enrichment analysis

All the biotinylated proteins enriched (with Enrichment score >1) in cells expressing Styxl2-BirA*-HA were imported and analysed at Geneontology (http://geneontology.org/). The terms in the cellular component category were presented.

## Immunoprecipitation

C2C12 cells or HEK293T cells were lysed in the lysis buffer with a cocktail of protease inhibitors. The soluble whole-cell lysates were first immunoprecipitated with an antibody targeting Styxl2, MEF2A (negative control), or Flag (Roche, F3165). The proteins co-precipitated were analysed by Western blot. In the ubiquitination assays, HEK 293T cells were lysed in the RIPA buffer (50 mM Tris-HCl, pH 7.4, 150 mM NaCl, 1 mM EDTA, 1% Triton X-100, 0.5% sodium deoxycholate, 0.1% SDS) with a cocktail of protease inhibitors and the soluble whole cell lysates were immunoprecipitated with an anti-Myc antibody (Santa Cruz, sc-40). The immunoprecipitated proteins were then analysed by Western blot.

## Injury-induced muscle regeneration

Avertin was used to anaesthetize mice at 0.5 mg per gram of body weight by intraperitoneal injection. Then 120 µl (for females) or 150 µl (for males) of 10 µM Cardiotoxin was injected into muscles of each hind limb to induce acute injury. After >30 days, the muscles were dissected and tested. To examine protein levels at different time points, each tibialis anterior muscle was injected with 30 µl of 1.2% BaCl$_2$ to induce acute damage. The regenerating muscles were collected for Western blot at different time points.

# Acknowledgements

The project was supported by research grants from the Hong Kong Research Grant Council (C6018-19G, AoE/M-604/16, T13-602/21 N, and T13-605/18 W), the State Key Laboratory of Molecular Neuroscience at HKUST, the Hong Kong Center for Neurodegenerative Diseases (HKCeND), and the Shenzhen Bay Laboratory (S201101002, Guangdong, China). We thank Drs. K Esser, M Rudnicki, and M Capecchi for providing mouse Cre lines used in this study. We also thank Dr. Y Guo for providing the BioID vector and Drs. Y Dai and S Zhao for their help with zebrafish work.

# Additional information

### Funding

| Funder | Grant reference number | Author |
| --- | --- | --- |
| Research Grants Council, University Grants Committee | C6018-19G | Zhenguo Wu |

| Funder | Grant reference number | Author |
|---|---|---|
| Research Grants Council, University Grants Committee | AoE/M-604/16 | Zhenguo Wu |
| Research Grants Council, University Grants Committee | T13-602/21-N | Zhenguo Wu |
| Research Grants Council, University Grants Committee | T13-605/18-W | Zhenguo Wu |
| Shenzhen Bay Laboratory | S201101002 | Zhenguo Wu |
| Innovation and Technology Commission | HKCeND | Zhenguo Wu |
| Innovation and Technology Commission | the State Key Laboratory of Molecular Neuroscience at HKUST | Zhenguo Wu |

The funders had no role in study design, data collection and interpretation, or the decision to submit the work for publication.

## Author contributions

Xianwei Chen, Conceptualization, Data curation, Formal analysis, Investigation, Methodology, Writing - original draft, Writing - review and editing; Yanfeng Li, Conceptualization, Data curation, Formal analysis, Investigation, Methodology; Jin Xu, Yong Cui, Qian Wu, Haidi Yin, Yuying Li, Investigation, Methodology; Chuan Gao, Data curation, Formal analysis, Investigation, Methodology; Liwen Jiang, Huating Wang, Zilong Wen, Zhongping Yao, Supervision, Investigation, Methodology; Zhenguo Wu, Conceptualization, Data curation, Formal analysis, Supervision, Funding acquisition, Investigation, Methodology, Writing - original draft, Project administration, Writing - review and editing

## Author ORCIDs

Xianwei Chen ⬦ http://orcid.org/0000-0002-5646-3245
Jin Xu ⬦ http://orcid.org/0000-0002-6840-1359
Huating Wang ⬦ http://orcid.org/0000-0001-5474-2905
Zhenguo Wu ⬦ http://orcid.org/0000-0003-3049-8324

## Ethics

All animals were maintained and handled following the protocols approved by the Animal Ethics Committee of the Hong Kong University of Science and Technology.

Reviewer #1 (Public Review): https://doi.org/10.7554/eLife.87434.3.sa1
Reviewer #2 (Public Review): https://doi.org/10.7554/eLife.87434.3.sa2
Reviewer #3 (Public Review): https://doi.org/10.7554/eLife.87434.3.sa3
Author response https://doi.org/10.7554/eLife.87434.3.sa4

# Additional files

## Supplementary files
• MDAR checklist

## Data availability

All data generated in this study are either included in the main figures and figure supplements or deposited to the Gene Expression Omnibus (accession number: GSE262242).Source data are also provided for the data presented in both main figures and figure supplements.

The following dataset was generated:

| Author(s) | Year | Dataset title | Dataset URL | Database and Identifier |
|---|---|---|---|---|
| Chen X | 2024 | Profiling of transcriptional targets of Jak1 and Stat1 in proliferating and differentiating myoblasts | https://www.ncbi.nlm.nih.gov/geo/query/acc.cgi?acc=GSE262242 | NCBI Gene Expression Omnibus, GSE262242 |

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
