## [Editor Report · eLife assessment]

This paper presents an **important** finding: that Styxl2, a poorly characterized pseudo-phosphatase, plays a role in the sarcomere assembly by promoting the degradation of non-muscle myosins. The genetic evidence supporting the conclusions is **compelling**, although future work will be needed to elucidate the functional role and biochemical mechanism of autophagic degradation of non-muscle myosins. This work will be of interest to biologists studying muscle development, cell biology, and proteolysis.

---

## [Referee Report · Reviewer #1 (Public Review)]

This paper performed a functional analysis of the poorly characterized pseudo-phosphatase Styxl2, one of the targets of the Jak/Stat pathway in muscle cells. The authors propose that Styxl2 is essential for de novo sarcomere assembly by regulating autophagic degradation of non-muscle myosin IIs (NM IIs). Although a previous study by Fero et al. (2014) has already reported that Styxl2 is essential for the integrity of sarcomeres, this study provides new mechanistic insights into the phenomenon. In vivo studies in this manuscript are compelling; however, I feel the contribution of autophagy in the degradation of NM IIs is still unclear.

---

## [Referee Report · Reviewer #2 (Public Review)]

The authors investigated the role of the Jak1-Stat1 signaling pathway in myogenic differentiation by screening the transcriptional targets of Jak1-Stat1 and identified Styxl2, a pseudophosphatase, as one of them. Styxl2 expression was induced in differentiating muscles. The authors used a zebrafish knockdown model and conditional knockout mouse models to show that Styxl2 is required for de novo sarcomere assembly but is dispensable for the maintenance of existing sarcomeres. Styxl2 interacts with the non-muscle myosin IIs, Myh9 and Myh10, and promotes the replacement of these non-muscle myosin IIs by muscle myosin IIs through inducing autophagic degradation of Myh9 and Myh10. This function is independent of its phosphatase domain.

A previous study using zebrafish found that Styxl2 (previously known as DUSP27) is expressed during embryonic muscle development and is crucial for sarcomere assembly, but its mechanism remains unknown. This paper provides important information on how Styxl2 mediates the replacement of non-muscle myosin with muscle myosin during differentiation. This study may also explain why autophagy deficiency in muscles and the heart causes sarcomere assembly defects in previous mouse models.

---

## [Referee Report · Reviewer #3 (Public Review)]

Wu and colleagues are characterising the function of Styxl2 during muscle development, a pseudo-phosphatase that was already described to have some function in sarcomere morphogenesis or maintenance (Fero et al. 2014). The authors verify a role for Styxl2 in sarcomere assembly/maintenance using zebrafish embryonic muscles by morpholino knock-down and by a conditional Styxl2 allele in mice (knocked-out in satellite cells with Pax7 Cre).

Experiments using a tamoxifen inducible Cre suggest that Styxl2 is dispensable for sarcomere maintenance and only needed for sarcomere assembly.

BioID experiments with Styxl2 in C2C 12 myoblasts suggest binding of nonmuscle myosins (NMs) to Styxl2. Interestingly, both NMs are downregulated when muscles differentiate after birth or during regeneration in mice. This down-regulation is reduced in the Styxl2 mutant mice, demonstrating that Styxl2 is required for the degradation of these NMs.

Impressively, reducing one NM (zMyh10) by double morpholino injection in a Styxl2 morphant zebrafish, does improve zebrafish mobility and sarcomere structure. Degradation of Mhy9 is also stimulated in cell culture if Styxl2 is co-expressed. Surprisingly, the phosphatase domain is not needed for these degradation and sarcomere structure rescue effects. Inhibitor experiments suggest that Styxl2 does promote the degradation of NMs by promoting the selective autophagy pathway.

Strengths:

A major strength of the paper is the combination of various systems, mouse and fish muscles in vivo to test Styxl2 function, and cell culture including a C2C12 muscle cell line to assay protein binding or protein degradation as well as inhibitor studies that can suggest biochemical pathways.

A second strength is that this manuscript sheds new light on the still ill-characterised mechanism of sarcomere assembly in skeletal muscles.

Weakness:

The weaknesses of this manuscript have been largely eliminated during revision.

---

## [Author Response]

The following is the authors’ response to the original reviews.

**Reviewer #1 (Public Review):**
This paper performed a functional analysis of the poorly characterized pseudo-phosphatase Styxl2, one of the targets of the Jak/Stat pathway in muscle cells. The authors propose that Styxl2 is essential for de novo sarcomere assembly by regulating autophagic degradation of non-muscle myosin IIs (NM IIs). Although a previous study by Fero et al. (2014) has already reported that Styxl2 is essential for the integrity of sarcomeres, this study provides new mechanistic insights into the phenomenon. In vivo studies in this manuscript are compelling; however, I feel the contribution of autophagy in the degradation of NM IIs is still unclear.Major concerns:1. The contribution of autophagy in the degradation of Myh9 is still unclear to this reviewer.It has been reported that autophagy is dispensable for sarcomere assembly in mice (Cell Metab, 2009, PMID; 1994508). In Fig. 7A, the authors showed that overexpressed Styxl2 downregulated the amount of ectopically expressed Myh9 in an ATG5-dependent manner in C2C12 cells; however, the experiment is far from a physiological condition. Therefore, the authors should test ATG5 knockdown and the genetic interaction between Styxl2 and ATG5 in vivo. That is, (1) loss of ATG5 on sarcomere assembly in zebrafish, and (2) the genetic interaction between Styxl2 and ATG5; co-injection of Styxl2 mRNA and ATG5-MO into the zebrafish embryos.

Our response: In fact, the reference cited by the reviewer (Cell Metab, 2009; PMID; 19945408) clearly indicated that autophagy is required for sarcomere assembly. Moreover, another paper using the fish extraocular muscle regeneration model (Autophagy, 2014, PMID: 27467399), also showed that the sarcomere structure was disrupted in the regenerated muscles when autophagy was inhibited by chloroquine. In addition, other references (Nature medicine, 2007, PMID: 17450150; Autophagy, 2010, PMID: 20431347) also showed that loss of Atg5 in mouse cardiac muscles led to disorganized sarcomere structure. We also performed the Atg5 knockdown experiments as suggested by the reviewer. However, the sarcomere structure defects were not so obvious as Styxl2 knockdown (see Author response image 1 below). In fact, it was reported that Atg5 knockdown may not be a desirable strategy to disrupt autophagy as it was found “--- only a small amount of Atg5 is needed for autophagy, knockdown of Atg5 to levels low enough to block autophagy might be difficult to achieve, --” (Nature medicine, 2007, PMID: 17450150). Due to the ineffectiveness of the Atg5 MO in our assays, we did not perform the second experiment suggested by the reviewer. Moreover, as Styxl2 is not a key component of the autophagy machinery, it is less likely that overexpression of Styxl2 alone can rescue the autophagy defects caused by Atg5.

**Author response image 1. sa4fig1:** The fish zygotes were injected with Atg5 or Ctrl MO. 48 hpf, the fish were stained with an anti-Actinin antibody. Some fast muscle fibers were disrupted when Atg5 was knocked down. The number in numerator at the bottom of each image represents fish embryos showing normal Actinin staining pattern, while that in denominator represents the total number of embryos examined. Scale bar, 10 µm.

1. As referenced, Yamamoto et al. reported that Myh9 is degraded by autophagy. Mechanistically, Nek9 acts as an autophagic adaptor that bridges Atg8 and Myh9 through interactions with both. Inconsistent with the model, the authors mentioned on page 12, lines 365-367, "A recent report showed that Myh9 could also undergo Nek9-mediated selective autophagy (Yamamoto et al., 2021), suggesting that Myh9 is ubiquitinated". I think it is not yet explored whether autophagic degradation of Myh9 requires its ubiquitination. Moreover, I cannot judge whether Myh9 is ubiquitinated in a Styxl2-dependent manner from the data in Fig. 7C. The author should test whether Nek9 is required for Myh9 degradation in muscles. If Nek plays a role in the Myh9 degradation, it would be better to remove Fig. 7C.

Our response: Indeed, as pointed out by the reviewer, it has not been explored whether Myh9 is ubiquitinated or not. However, it has been well-established that some proteins undergoing autophagic degradation are ubiquitinated, which are linked to Atg8/LC3 via p62 and NBR1 (Mol Cell, 2009, PMID: 19250911; J Biol Chem, 2007, PMID: 17580304). To improve the data quality, we repeated the Myh9 ubiquitination experiment in cells with or without Styxl2 by using a slightly different strategy: as shown in the revised Figure 7C, we first co-transfect HEK 293T cells with HA-Myh9, Myc-ubiquitin, and Flag-Styxl2. We then immunoprecipitated Myc-tagged Ubiquitin from the whole cell lysates, and then blot for HAMyh9. We detected an obvious increase in Ubiquitin-conjugated HA-Myh9 (revised Figure 7C). As suggested by the reviewer, we also tested whether knockdown of Nek9 affects the degradation of Myh9. We failed to detect an obvious effect (see Author response image 2 below) caused by Nek9 knockdown. One possible explanation for this negative result is that Nek9 itself is a negative regulator of selective autophagy (J Biol Chem, 2020, PMID: 31857374). By knocking it down, the functions of the autophagy machinery are expected to be enhanced instead of being impaired. This may explain why we failed to detect an effect on Myh9 degradation simply by knocking down Nek9. To further elucidate whether Nek9 is involved in Myh9 degradation in myoblasts, we may need to use a dominant-negative mutant of Nek9 missing the LCIII-binding motif as shown by Yamamoto (Nat Commun, 2021, PMID:34078910). This will be addressed in our future study.

**Author response image 2. sa4fig2:** C2C12 cells were transfected with negative control siRNA (NC), siNek9#2 or siNek9#3. 18 h later, the cells were transfected with plasmids HA-Myh9 and Flag-Styxl2 or Flag-Stk24. After another 24 h, the cells were harvested for RT-qPCR (left panel) or western blot (right panel).

1. In Fig. 5F, the protein level of Styxl2 and Myh10 should be checked because the efficiency of Myh10-MO was not shown anywhere in this manuscript.

Our response: As suggested by the reviewer, a Western blot showing the protein levels of Myh10 was shown in Figure 5-figure supplement 1B.

**Reviewer #2 (Public Review):**
The authors investigated the role of the Jak1-Stat1 signaling pathway in myogenic differentiation by screening the transcriptional targets of Jak1-Stat1 and identified Styxl2, a pseudophosphatase, as one of them. Styxl2 expression was induced in differentiating muscles. The authors used a zebrafish knockdown model and conditional knockout mouse models to show that Styxl2 is required for de novo sarcomere assembly but is dispensable for the maintenance of existing sarcomeres. Styxl2 interacts with the non-muscle myosin IIs, Myh9 and Myh10, and promotes the replacement of these non-muscle myosin IIs by muscle myosin IIs through inducing autophagic degradation of Myh9 and Myh10. This function is independent of its phosphatase domain.A previous study using zebrafish found that Styxl2 (previously known as DUSP27) is expressed during embryonic muscle development and is crucial for sarcomere assembly, but its mechanism remains unknown. This paper provides important information on how Styxl2 mediates the replacement of non-muscle myosin with muscle myosin during differentiation. This study may also explain why autophagy deficiency in muscles and the heart causes sarcomere assembly defects in previous mouse models.
**Reviewer #3 (Public Review):**
Wu and colleagues are characterising the function of Styxl2 during muscle development, a pseudo-phosphatase that was already described to have some function in sarcomere morphogenesis or maintenance (Fero et al. 2014). The authors verify a role for Styxl2 in sarcomere assembly/maintenance using zebrafish embryonic muscles by morpholino knockdown and by a conditional Styxl2 allele in mice (knocked-out in satellite cells with Pax7 Cre).Experiments using a tamoxifen inducible Cre suggest that Styxl2 is dispensable for sarcomere maintenance and only needed for sarcomere assembly.BioID experiments with Styxl2 in C2C 12 myoblasts suggest binding of nonmuscle myosins (NMs) to Styxl2. Interestingly, both NMs are downregulated when muscles differentiate after birth or during regeneration in mice. This down-regulation is reduced in the Styxl2 mutant mice, suggesting that Styxl2 is required for the degradation of these NMs.Impressively, reducing one NM (zMyh10) by double morpholino injection in a Styxl2 morphant zebrafish, does improve zebrafish mobility and sarcomere structure. Degradation of Mhy9 is also stimulated in cell culture if Styxl2 is co-expressed. Surprisingly, the phosphatase domain is not needed for these degradation and sarcomere structure rescue effects. Inhibitor experiments suggest that Styxl2 does promote the degradation of NMs by promoting the selective autophagy pathway.Strengths:A major strength of the paper is the combination of various systems, mouse and fish muscles in vivo to test Styxl2 function, and cell culture including a C2C12 muscle cell line to assay protein binding or protein degradation as well as inhibitor studies that can suggest biochemical pathways.Weakness:The weakness of this manuscript is that the sarcomere phenotypes and also the western blots are not quantified. Hence, we rely on judging the results from a single image or blot. Also, Styxl2 role in sarcomere biology was not entirely novel.Few high resolution sarcomere images are shown, myosins have not been stained for.
**Reviewer #1 (Recommendations For The Authors):**
Minor concerns:1. The position of molecular weight markers should be shown in all Western blot data.

Our response: As suggested by the reviewer, the molecular weight markers have been added in the Western blot data.

1. Schematic models of Styxl2deltaN509 and N513 construct would be helpful for the readers.

Our response: A schematic has been added in Figure 6B (upper panel) to show Styxl2deltaN509 and Styxl2N513.

1. Several data were described but not shown (data not shown). I think the data need to be included in the main or supplemental figures.

Our response: As suggested by the reviewer, the raw data were now included in the Figure 6-figure supplement 1A and Figure 7-figure supplement 1.

**Reviewer #2 (Recommendations For The Authors):**
1. In Fig. 5E, the authors suggest that the needle touch response was improved by additional knockdown of Myh10. This is a bit confusing because the germline knockout of Myh10 is lethal (line 445). The authors should provide more explanation on this point. Additionally, it would be better to include Myh10-MO in Fig. 5E.

Our response:

In line 445 of our original manuscript, we stated that germline knockout of mouse Myh10 gene is lethal based on a published report (Proc Natl Acad Sci USA, 1997, PMID: 9356462). Here, in zebrafish zygotes, we only knocked down zMyh10, thus, we do not expect to get a lethal phenotype. In addition, other groups who knocked down Myh10 in fish also did not get a lethal phenotype (Dev Biol, 2015, PMID: 25446029). As to the control involving Myh10MO in the experiment in Fig.5E, we did include it in our experiments. As we did not observe any obvious effects on either motility or sarcomere structures, we did not include the data set in the figure.

1. It was suggested that Myh9 and Myh10 form a complex (Rao et al. PLoS One 9, e114087, 2014). Thus, the IP experiments do not rule out the possibility that Styxl2 directly interacts with either Myh9 or Myh10 and indirectly with the other.

Our response: In known myosin-II complexes, different myosin molecules can associate with each other through their tail domains (Bioarchitecture, 2013, PMID: 24002531). Thus, if we use fulllength myosin molecules in our co-immunoprecipitation assays, it will be difficult to exclude the possibility raised by the reviewer. However, by using truncated myosin proteins, we showed that the head domain of either Myh9 or Myh10 could interact with Styxl2 in the absence of the tail domain (Figure 4E, F). This result strongly suggests that both Myh9 and Myh10 can independently interact with Styxl2.

**Reviewer #3 (Recommendations For The Authors):**
1. The western blot shown in Figure 3B supporting the induced deletion of Styxl2 should be quantified. Ideally, some other blots, e.g., in Figure 5, too. Please add the age of the mice in Figure 5B to the figure legend.

Our response:

As suggested by the reviewer, we quantified the data in Figures.3B, 3F, 5B, 5D, and 7A and the data were included in the revised figures. In Fig.5B, we already indicated the age of the mice (i.e., P1) in the legend.

1. A quantification of the sarcomere phenotypes in the double knock-down of zMyh10 and Styxl2 compared to Styxl2 single would make the paper significantly stronger. Furthermore, a double morpholino control should be included to rule out any RNAi machinery 'dilution effect'.

Our response: As suggested by the reviewer, we quantified the sarcomere structures using the line scan analysis in ImageJ and the scan images were placed as inserts in the upper corner of the immunofluorescent images (revised Figures 5F, and 6C). To avoid potential “dilution effects”, in all the experiments involving the use of two different MOs, the total amount of MO was kept the same in all control samples by including a control MO (e.g., in samples treated with one specific MO, an equal amount of a control MO was also included, while in samples without any specific MO, twice as much control MO was used).

1. The sarcomere phenotypes in figure 6 should also be better quantified, for example using simple line scans of the alpha-actinin stains and assay periodicity or calculating the autocorrelation coefficients. How about myosin stains?

Our response: We quantified Figure 6C as suggested by the reviewer. We also performed myosin staining. The results were similar to that shown by the a-actinin antibody (see revised Figure 6-Fig supplement 1B).

1. Do the authors see periodic NMs patterns in developing mouse muscle fibers as indicated by the model in in in figure 7D? It is unclear if nonmuscle myosin is present in a PERIODIC pattern in early myofibrils. NM myosin periodic patterns that have been observed have a periodicity of only about 1 µm fitting the shorter length of the NM bipolar filaments (about 300 nm only, PMID 28114270).

Our response: The reviewer raised a good point here. Ideally, we should examine developing mouse muscle fibers to prove that NM shows periodic patterns. However, due to the difficulty in catching myocytes undergoing sarcomere assembly, the majority of the studies involving NM in sarcomeres use cultured cardiomyocytes. Using TA muscles from P1 new-born mice, we failed to detect the presence of NM in sarcomeres (see Author response image 3 below). Actually, nearly all the myofibers showed mature sarcomere pattern without the NM signal. More work is needed in the future to examine developing mouse fibers at different embryonic stages to look for the presence of NM in developing sarcomeres.

**Author response image 3. sa4fig3:** The TA muscles were collected from male and female P1 mice. The muscles were sectioned and co-stained for a-actinin (Actn) and Myh9. The majority of myofibrils is mature without the NM II signal. Scale bar, 10 µm.

1. Recent work suggested that mechanical tension is key to assemble the first long periodic myofibril containing immature sarcomeres. Tension is likely produced by a combination of NM and Mhc in the assembling sarcomeres themselves. This could be included in the introduction or discussion (PMIDs 24631244, 29316444, 29702642, 35920628).

Our response: We thank the reviewer for pointing to us additional relevant references. We have added them in the Introduction.

1. I suggest replacing "sarcomeric muscles" with "striated muscles".

Our response: We revised the term in the manuscript as suggested by the reviewer.